# Burden of hemoglobinopathies and hemolytic anemias in the World Health Organization African region, 2000–2021: Findings from the Global Burden of Disease 2021 study

GBD 2021 Hemoglobinopathies and Hemolytic Anemias Collaborators, Temitope T Ojo[1], Prince M Amegbor[2], Farha Islam[3], Joyce Gyamfi[2], Andi Mai[4], Carly M Malburg[2], Deborah B Adenikinju[2], Nicholas J Kassebaum[5,6,7], Shimelis Tadesse Abebe[8], Richard Gyan Aboagye[9], Ganiyu Adeniyi Amusa[10,11], Seth Christopher Yaw Appiah[12,13], Haftu Asmerom Asmerom[14], Isaac Sunday Chukwu[15], Tadesse Asmamaw Dejenie[16], Fitsum Wolde Demisse[17], Gashaw Dessie[18], Mengistie Diress[19], Christopher Imokhuede Esezobor[20,21], Habitu Birhan Eshetu[22], Adeniyi Francis Fagbamigbe[23,24], Sefineh Fenta[25], Teferi Gebru Gebremeskel[26,27], Segun Emmanuel Ibitoye[28], Robel Hussen Kabthymer[29], Woldeteklehaymanot Dagne Kassahun[30], Biruk Getahun Kibret[31], Osaretin Christabel Okonji[32], Prof Mayowa O Owolabi[33,34], Prof Léon Muepu M Tshilolo[35,36], Berhanu Woldu[37], Emmanuel K Peprah[2]*

1 Department of Social and Behavioral Sciences, New York University, New York, New York, United States of America, 2 School of Global Public Health, New York University, New York, New York, United States of America, 3 New York University, New York, New York, United States of America, 4 Epidemiology and Biostatistics Department, Indiana University Bloomington, Bloomington, Indiana, United States of America, 5 Department of Anesthesiology & Pain Medicine, University of Washington, Seattle, Washington, United States of America, 6 Institute for Health Metrics and Evaluation, University of Washington, Seattle, Washington, United States of America, 7 Department of Health Metrics Sciences, School of Medicine, University of Washington, Seattle, Washington, United States of America, 8 Department of Midwifery, Mattu University, Mattu, Ethiopia, 9 Department of Family and Community Health, University of Health and Allied Sciences, Ho, Ghana, 10 Department of Medicine, University of Jos, Jos, Nigeria, 11 Department of Internal Medicine, Jos University Teaching Hospital, Jos, Nigeria, 12 Department of Sociology and Social Work, Kwame Nkrumah University of Science and Technology, Kumasi, Ghana, 13 Center for International Health, Ludwig Maximilians University, Munich, Germany, 14 School of Medical Laboratory Sciences, Haramaya University, Harar, Ethiopia, 15 Department of Paediatric Surgery, Federal Medical Centre, Umuahia, Nigeria, 16 Department of Medical Biochemistry, University of Gondar, Gondar, Ethiopia, 17 Department of Midwifery, Arba Minch University, Arba Minch, Ethiopia, 18 Biochemistry Department, University of Gondar, Gondar, Ethiopia, 19 Department of Human Physiology, University of Gondar, Gondar, Ethiopia, 20 Department of Paediatrics, University of Lagos, Lagos, Nigeria, 21 Department of Paediatrics, Lagos University Teaching Hospital, Lagos, Nigeria, 22 Department of Health Promotion and Health Behavior, University of Gondar, Gondar, Ethiopia, 23 Department of Epidemiology and Medical Statistics, University of Ibadan, Ibadan, Nigeria, 24 Research Centre for Healthcare and Community, Coventry University, Coventry, United Kingdom, 25 Department of Public Health, Woldia University, Woldia, Ethiopia, 26 Department of Reproductive and Family Health, Axum College of Health Science, Axum, Ethiopia, 27 College of Medicine and Public Health, Flinders University, Adelaide, South Australia, Australia, 28 Department of Health Promotion and Education, University of Ibadan, Ibadan, Nigeria, 29 School of Public Health, Dilla University, Dilla, Ethiopia, 30 Department of Medical Laboratory Sciences, Woldia University, Woldia, Ethiopia, 31 Department of Medical Physiology, Bahir Dar University, Bahir Dar, Ethiopia, 32 School of Pharmacy, University of the Western Cape, Cape Town, South Africa, 33 Department of Medicine, University of Ibadan, Ibadan, Nigeria, 34 Department of Medicine, University College Hospital, Ibadan, Nigeria, 35 Institut de Recherche Biomédicale (IRB), Centre de Formation et d'Appui Sanitaire "CEFA-Monkole", Kinshasa, Democratic Republic of the Congo, 36 Département de Pédiatrie, Faculté de Médecine, Université Officielle de Mbujimayi "UOM", Mbujimayi, Democratic Republic of the Congo, 37 Department of Hematology and Immunohematology, University of Gondar, Gondar, Ethiopia

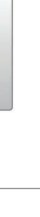

**Data availability statement:** The GBD HHA data is available here https://ghdx.healthdata.org/gbd-2021.

**Funding:** The author(s) received no specific funding for this work.

**Competing interests:** L M M Tshilolo reports grants or contracts from Research Evaluation and Commercialization Hubs (REACH)/National Heart, Lung, and Blood Institute (NIHLB) for a study on the use of hydroxyurea in Sub-Saharan African countries, and the SickleInAfrica consortium; consulting fees from Novo Nordisk and Novartis; payment or honoraria for lectures, presentations, speakers bureaus, manuscript writing or educational events from Novo Nordisk for the AS FacultyHeroes Course; support for attending meetings and/or travel from Novo Nordisk and WHO AFRO; leadership or fiduciary roles in other board, society, committee or advocacy groups, paid or unpaid, with REDAC as President, and the SickleInAfrica consortium as Co-chair; other financial or non-financial interests with the Belgian Development Cooperation (DGD) Association for Cultural, Technical and Educational Cooperation (ACTEC) via an education project for bio-technicians in the Democratic Republic of the Congo; all outside the submitted work.

* ep91@nyu.edu, dr.epeprah@gmail.com

## Abstract

Hemoglobinopathies and hemolytic anemias (HHA) are genetic blood disorders associated with diverse clinical complications, affecting an estimated 2.1 billion people worldwide. The World Health Organization (WHO) African Region accounts for approximately 425.8 million individuals, or 20% of the global HHA prevalence, yet comprehensive assessments of this burden have been lacking. We present the first systematic analysis of HHA burden in the WHO African Region from 2000–2021 using data from the Global Burden of Disease (GBD) 2021 study. We estimated regional, sex-, and age-specific rates (per 100,000 population) of mortality, incidence at birth, and years lived with disability (YLDs) in five-year intervals. Mortality estimates were generated using the Cause of Death Ensemble model (CODEm), supplemented with spatiotemporal Gaussian process regression. Incidence at birth was estimated using DisMod-MR 2.1, a Bayesian meta-regression tool, while YLDs were calculated by multiplying prevalence by disability weights reflecting severity and duration. Between 2000 and 2021, the WHO African Region experienced persistently higher age-standardized death rates from HHA compared to global levels, although regional mortality declined over the period. Sickle cell disorder (SCD) was the predominant contributor, with the highest mortality [3.68 deaths (95% UI 2.04–6.29) per 100,000] and disability burden [41.08 YLDs (95% UI 26.09–58.61)], while thalassemias contributed the least. Disability-adjusted life years (DALYs) were concentrated in western sub-Saharan Africa, accounting for 71.3% of the regional burden. Age-specific estimates revealed that children under five years faced a disproportionate share of mortality and disability. Despite overall declines in mortality, the WHO African Region continues to bear a disproportionate global burden of HHA, particularly affecting young children. These findings underscore the urgent need for strengthened newborn screening, early treatment, and health system interventions to reduce preventable deaths and disability.

### Introduction

Hemoglobinopathies and hemolytic anemias (HHA) are a group of genetic blood disorders that, when inherited in both chromosomes, results in various clinical pathologies to erythrocytes [1–3]. These pathologies result in a spectrum of clinical manifestations, such as acute pain crises, anemia, vaso-occlusive events, acute chest syndrome, fatigue, hemolysis, and jaundice, when individuals inherit both recessive genes of these blood disorders. HHA are most endemic in sub-Saharan Africa, the Mediterranean, the Middle East, and the Indian subcontinent. The WHO African

Region carries a disproportionate burden of HHA, with the most regional deaths associated with these blood disorders occurring in sub-Saharan Africa [4].

According to the Global Burden of Disease (GBD) 2019 study, the global prevalence of HHA is approximately 2.1 billion people, with the WHO African Region accounting for 20% (approximately 425.8 million people) of this global prevalence. Moreover, a recent analysis of GBD estimates 515 000 births from SCD [5] globally, however, a comprehensive analysis of the HHA burden, including incidence-at-birth, disability-adjusted life-years (DALYs), and years lived with disability (YLDs), has not been conducted for the WHO African Region. As of 2021, 49% (43,153) of global deaths due to HHA were from the WHO African Region [6]. Children under the age of 5 are impacted more than any other age group by HHA-specific deaths and disability [7,8]. Of the seven categories of HHA identified by the GBD studies, SCD has the most severe pathophysiology, resulting in painful crises characterized by extreme pain and lower life expectancy compared to G6PD and thalassemias [1,9]. G6PD deficiency affects over 400 million people and is the most common enzymopathy worldwide with a prevalence 15–26% in African populations [10–12]. Thalassemias are the most common single gene disorders around the globe [13]. Alpha (α)-thalassemia affects about 5% of the world's population [14] in contrast, beta (β)-thalassemia which requires lifelong blood transfusions and iron chelation therapy to prevent complications from iron overload is estimated to affect about 1.5% of world population (80–90 million people) [15]. Nevertheless, among HHA sickle-cell disorders was responsible for 80% of haemoglobinopathy-related deaths in the WHO African Region; 1–3% of African populations influenced by sickle-cell disorder [16,17]. Despite advancements in evidence-based clinical innovations including diagnostics and therapies to manage clinical manifestations and improve health outcomes for SCD(e.g., newborn screening, health education, optimal nutrition, hydroxyurea therapy, blood transfusion, prophylaxis for infections, and bone-marrow transplantation) many of these innovations are underutilized in the WHO African Region which still bears the greatest burden of SCD, as well as HHA in general [18,19].

This aligns with documented evidence that a majority of HHA-affected populations in Africa face challenges in accessing and affording these evidence-based interventions (EBIs) to manage their condition [20–22]. This study presents results on HHA burden in the WHO African Region for 22 years (2000–2021), provides time trends for HHA burden in the region, and highlights significant changes in the burden over time. Moreover, a careful examination of the data provides an opportunity to observe possible gains in prevention and management of HHA over time in Africa, through the temporal change in the HHA burden profile.

## Methods

The GBD studies use a comprehensive and diverse database (published and unpublished), derived from censuses, administrative health records, vital registrations, disease registries, surveys, medical claims, scientific literature, verbal autopsy studies, and disease surveillance systems from different countries [23,24]. GBD 2021 methodology has been published previously [23–28] and uses various statistical modelling techniques, the GBD database provides estimates of disease measures, imputing estimates for countries and regions where data are scarcely available. This was described in greater detail in an earlier GBD study publication [23–25]. GBD estimates are available as age-standardized annual mortality rates per 100,000 people, number and percentages of cause-specific deaths, prevalence, incidence, DALYs, YLLs, and YLDs, with corresponding uncertainty levels, trends, and changes from 2000 to 2021 [23,24]. Each metric is accompanied by a 95% uncertainty interval (UI), which is derived from the 2.5th and 97.5th ordered values of 1,000 posterior simulations [29]. Disability weights used in YLD estimation are generated from large-scale population-based surveys, designed to capture community perceptions of disease severity and functional health loss. Estimates were stratified by age groups, sex, and geographical region. Publicly accessible GBD estimates and metadata can be explored through the GBD Results Tool, which provides country-level and regional estimates across the full range of available metrics [6].

In this study, we focused on hemoglobinopathies and hemolytic anemias (HHA) in 47 countries within the WHO African Region. We report on sex-specific, age-specific, and regional-level annual rates per 100,000 population

for three main outcomes: Cause-specific mortality, Incidence at birth, DALYs and YLDs from 2000 to 2021, in five-year intervals. Regional estimates were derived from the GBD super-region of sub-Saharan Africa, reported as four sub-categories, and GBD region of Africa under the "Four World Regions" GBD categories. To reflect the inherent uncertainty in the data, all metrics reported incorporate GBD's simulation-based uncertainty modeling. This ensures that even in countries with limited direct data availability, results are statistically valid and consistent with observed regional trends [30]. We also presented higher-level estimates and time trends of HHA in percentages and numbers. Cause-specific mortality due to HHA was modeled using the Cause of Death Ensemble model (CODEm), a robust tool that evaluates and integrates multiple predictive models to produce the most accurate mortality estimates. This was further supplemented with spatiotemporal Gaussian process regression to account for geographic and temporal variability in the data. To calculate YLLs, the number of deaths at each age was multiplied by the standard life expectancy for that age, based on the GBD reference life table [31]. The incidence at birth and prevalence of HHAs were estimated using DisMod-MR 2.1, a Bayesian meta-regression tool that synthesizes data from diverse sources, accounting for variations in case definitions, data quality, and reporting. This model ensures internal consistency across disease parameters (incidence, prevalence, remission, and mortality). YLDs were calculated by multiplying age- and sex-specific prevalence estimates by corresponding disability weights. These weights quantify the non-fatal health loss attributable to each condition, on a scale from 0 (perfect health) to 1 (death). DALYs were computed as the sum of YLDs and YLLs, capturing both premature mortality and morbidity in a single metric [29]. To provide further insight into the relative burden of HHA-related mortality, we calculated the standardized mortality ratio (SMR) for HHA across age groups and time periods. The SMR expresses HHA deaths as a proportion of total deaths, allowing comparison of HHA's share in overall mortality across different demographic groups. The GBD HHA data is available here https://ghdx.healthdata.org/gbd-2021.

## Results

Compared to global estimates of death due to HHA, the WHO-African region has experienced higher age-standardized death rates due to HHA from 2000 to 2021(see Fig 1A & 1B). Nevertheless, the 22-year span showed that WHO-African region's age-standardized death rate due to HHA diminished from 2000 to 2021 (see Fig 1B and Tables 1,2). Of the sub-categories of HHA assessed in 2000 (i.e., G6PD, SCD, thalassemias) in the WHO Africa Region (Table 3), age-standardized death rate for G6PD was 20.06 (95% UI 13.57 – 34.32), SCD caused.55 deaths (95% UI 0.33 – 1.03) and thalassemias was.28 (95% UI 0.19 – 0.41), per 100,000 population. By 2021, the age-standardized death rate had decreased for G6PD to 17.49 (95% UI 10.90 – 30.61), or was fairly consistent for SCD 0.52 deaths (95% UI 0.31 – 0.96) and thalassemias caused 0.21 (95% UI 0.14 – 0.30), per 100,000 population compared to 2000.

HHA incidence at birth (per 1000 live births) in the WHO African Region has been stable over the 22-year span, with rates of 602.29 (95% UI, 576.55 - 628.69) in 2000 and 596.79 (95% UI, 570.25 - 625.25) in 2021, despite slight increases in 2008 and 2012 (Table 5). Sickle cell trait had the highest incidence at birth for both males and females, remaining consistent over the 22-year period. Furthermore, the male-to-female birth incidence ratios for HHA sub-types, including SCD, thalassemias, and G6PD, have remained relatively unchanged over time.

### Regional estimates

In 2000, WHO Africa regional specific age-standardized deaths were highest for Western Sub-Saharan Africa, accounting for 23,480 (95% UI, 14,158–35,343) deaths. By 2021, the number of deaths had increased in all regions, with the Western Sub-Saharan region experiencing the highest increase of 27.7% within the WHO Africa region, resulting in a total of 31,213 (95% UI: 19,743–47,900) deaths in 2021 (Table 1; Fig 2). Regional cause-specific age-standardized rate of deaths from HHA were highest for Western Sub-Saharan Africa in 2000 and decreased over the 22-year span from 8.76

A

### Comparison of WHO-African Region to Global age-standardised death rate due to HHA from 2000 to 2021

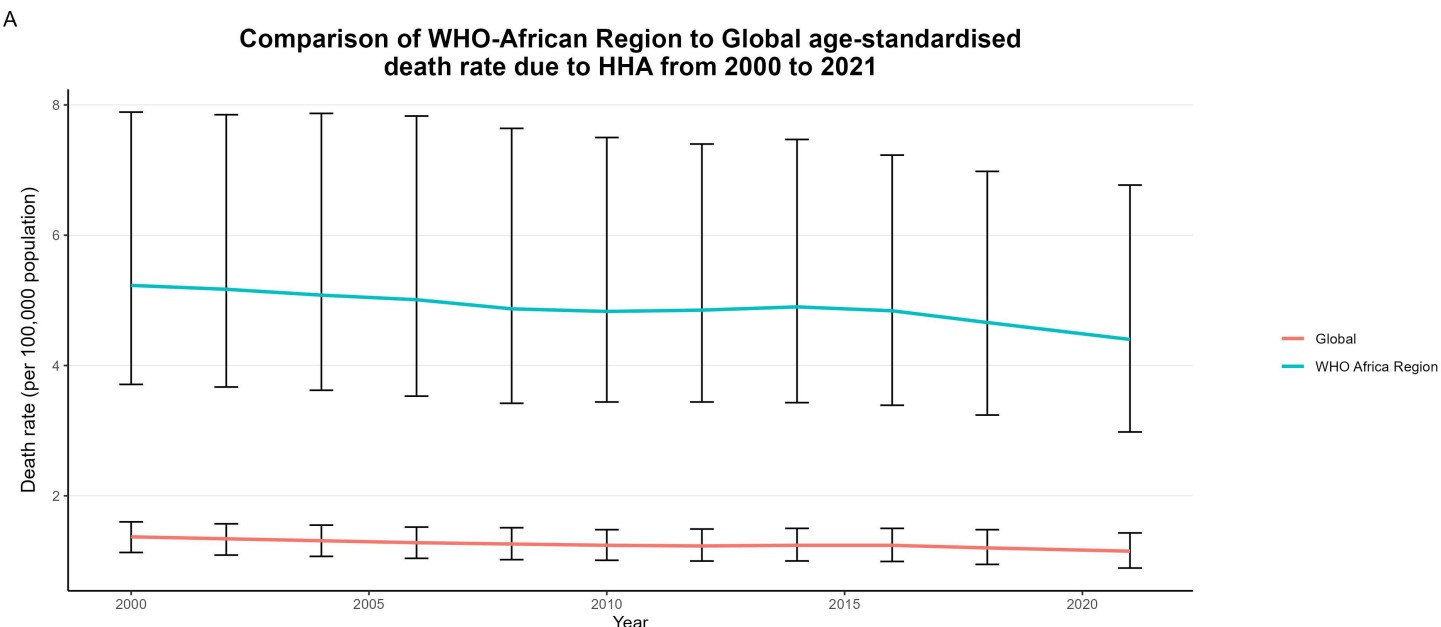

B

### WHO Africa Region age-standardized death rate by HHA subtype, 2000 - 2021

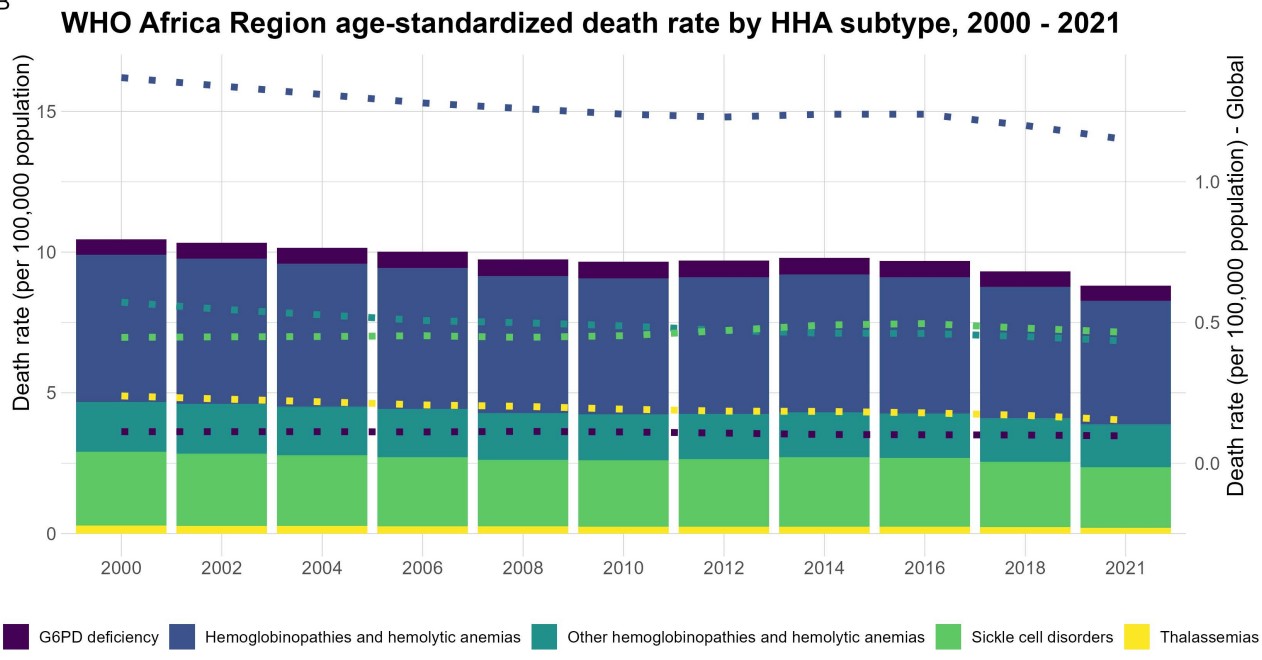

Legend: G6PD deficiency · Hemoglobinopathies and hemolytic anemias · Other hemoglobinopathies and hemolytic anemias · Sickle cell disorders · Thalassemias

**Fig 1. HHA age-standardized death rates: WHO Africa vs. global and by subtype (2001–2021). A**, Comparison of WHO-African region to global age-standardised death rate due to HHA from 2000 to 2021. **B**, WHO Africa Region Age-standardised death rate by HHA subtype, 2000–2021. The global age-standardized death rate is represented as a line graph for comparison.

(95% UI: 5.61 – 12.47) to 6.97(95% UI: 4.54 – 10.07) (Table 2). Southern Sub-Saharan Africa experienced the smallest decrease in age-standardized death rates from 2000 to 2021, followed by Eastern Sub-Saharan Africa. However, Central Sub-Saharan Africa has now surpassed Western Sub-Saharan Africa as the region with the highest death rates (7.25;

**Table 1. Regional cause-specific deaths (in numbers) due to haemoglobinopathies in Sub-Saharan Africa, from 2000 to 2021.**

| Region | 2000 | 2001 | 2002 | 2003 | 2004 | 2005 | 2006 | 2007 | 2008 | 2009 | 2010 |
|---|---|---|---|---|---|---|---|---|---|---|---|
| WHO Africa Region | 32,213 (20,431 - 50,201) | 32,737 (20,856 - 51,139) | 33,311 (21,324 - 51,815) | 33,746 (21,641 - 52,319) | 34,194 (22,159 - 53,470) | 34,638 (22,556 - 54,466) | 35,352 (22,918 - 55,065) | 35,767 (23,418 - 55,779) | 35,930 (23,422 - 55,818) | 36,655 (23,993 - 56,800) | 37,504 (24,652 - 57,628) |
| Western Sub-Saharan Africa | 23,480 (14,158 - 35,343) | 23,862 (14,466 - 36,038) | 24249 (14,901 - 36,436) | 24,437 (14,939 - 36,693) | 24,706 (15,497 - 37,264) | 24,977 (15,690 - 37,327) | 25,476 (15,948 - 38,004) | 25,739 (16,202 - 37,966) | 25,725 (16,189 - 37,997) | 26,254 (16,540 - 38,818) | 26,926 (17,050 - 39,640) |
| Central Sub-Saharan Africa | 3,819 (764 - 11,506) | 3,915 (786 - 11,962) | 4,010 (800 - 12,381) | 4,145 (830 - 12,823) | 4,205 (844 - 13,068) | 4,287 (864 - 13,377) | 4,401 (896 - 13,837) | 4,476 (923 - 13,957) | 4,573 (954 - 14,336) | 4,688 (986 - 14,576) | 4,802 (1,022 - 15,034) |
| Eastern Sub-Saharan Africa | 4,059 (2,099 - 7,310) | 4,076 (2,139 - 7,293) | 4,135 (2,197 - 7,460) | 4,219 (2,250 - 7659) | 4,328 (2,326 - 7,881) | 4,408 (2,370 - 8,037) | 4,505 (2,433 - 8,242) | 4,603 (2,488 - 8,455) | 4,666 (2,531 - 8,547) | 4,746 (2,575 - 8,785) | 4,813 (2,618 - 8,873) |
| Southern Sub-Saharan Africa | 741 (594 - 909) | 776 (628 - 954) | 819 (667 - 1,020) | 859 (701 - 1,082) | 881 (719 - 1,117) | 901 (733 - 1,132) | 913 (740 - 1,143) | 905 (734 - 1,126) | 938 (755 - 1,169) | 947 (758 - 1,187) | 952 (754 - 1,188) |
| North Africa & Middle East | 3,923 (3,068 - 5,084) | 3,893 (3,045 - 5,024) | 3,879 (3,039 - 4,996) | 3,887 (3,054 - 5,007) | 3,906 (3,078 - 5,033) | 3,920 (3,108 - 5,041) | 3,923 (3,126 - 5,024) | 3,927 (3,135 - 5,023) | 3,944 (3,170 - 5,049) | 3,962 (3,209 - 5,049) | 3,943 (3,219 - 5,011) |
| Global | 76,872 (57,760 - 99,321) | 76,995 (57,909 - 99,500) | 77,264 (58,162 - 99,779) | 77,577 (58,721 - 100,117) | 77,938 (59,269 - 100,844) | 78,454 (59,902 - 101,695) | 78,858 (59,729 - 102,055) | 79,542 (60,666 - 103,256) | 80,330 (61,131 - 103,742) | 81,082 (61,789 - 104,485) | 81,853 (62,723 - 105,390) |

95% UI in parentheses.

| 2011 | 2012 | 2013 | 2014 | 2015 | 2016 | 2017 | 2018 | 2019 | 2020 | 2021 |
|---|---|---|---|---|---|---|---|---|---|---|
| 38,486 (25,347 - 59,628) | 39,827 (26,390 - 61,115) | 41,292 (27,225 - 63,809) | 42,612 (28,165 - 65,528) | 43,374 (28,671 - 66,527) | 44,209 (29,388 - 67,325) | 44,234 (29,281 - 67,140) | 43,895 (29,037 - 67,483) | 43,593 (28,849 - 67,071) | 43,481 (28,218 - 67,720) | 43,153 (27,978 - 67,646) |
| 27,695 (17,353 - 40,898) | 28,819 (18,093 - 42,206) | 30,060 (18,808 - 44,437) | 31,132 (19,521 - 46,340) | 31,714 (19,967 - 46,773) | 32,418 (20,500 - 47,812) | 32,409 (20,524 - 48,805) | 32,019 (20,305 - 48,387) | 31,683 (20,152 - 47,474) | 31,514 (19,735 - 47,466) | 31,213 (19,743 - 47,900) |
| 4,930 (1,055 - 15,361) | 5,086 (1,111 - 15,733) | 5,245 (1,156 - 16,216) | 5,411 (1,205 - 16,760) | 5,507 (1,240 - 17,175) | 5,564 (1,256 - 17,512) | 5,588 (1,273 - 17,541) | 5,630 (1,282 - 17,795) | 5,643 (1,294 - 17,658) | 5,692 (1,308 - 17,993) | 5,700 (1,311 - 18,053) |
| 4,895 (2,673 - 9,052) | 4,954 (2,723 - 9,199) | 5,032 (2,774 - 9,350) | 5,110 (2,834 - 9,530) | 5,178 (2,867 - 9,672) | 5,239 (2,935 - 9,889) | 5,259 (2,937 - 9,952) | 5,252 (2,984 - 10,032) | 5,272 (3,020 - 10,114) | 5,281 (3,022 - 10,204) | 5,239 (3,003456789 - 10,170) |
| 962 (756 - 1,199) | 965 (755 - 1,209) | 954 (744 - 1,193) | 959 (741 - 1,205) | 974 (753 - 1,232) | 996 (767 - 1,265) | 989 (757 - 1,263) | 1,008 (777 - 1,290) | 1015 (781 - 1,307) | 1,022 (787 - 1,312) | 1,027 (786 - 1,328) |
| 3,920 (3,216 - 4,974) | 3,903 (3,204 - 4,944) | 3,925 (3,227 - 4,961) | 3,919 (3,227 - 4,961) | 3,880 (3,202 - 4,923) | 3,823 (3,157 - 4,846) | 3,751 (3,094 - 4,755) | 3,696 (3,040 - 4,716) | 3,683 (3,015 - 4,737) | 3,593 (2,923 - 4,650) | 3,542 (2,868 - 4,613) |
| 82,703 (63,360 - 107,162) | 83,846 (64,177 - 108,358) | 85,351 (65,231 - 111,103) | 86,849 (66,495 - 113,160) | 87,891 (67,216 - 115, 050) | 89,291 (68,389 - 115,826) | 89,431 (68,089 - 116,439) | 89,122 (67,765 - 116,185) | 88,950 (67,548 - 116, 002) | 88,445 (66,570 - 116,466) | 88,121 (66,406 - 115,663) |

**Table 2. Regional cause-specific age-standardised rate of deaths (per 100,000 population) due to haemoglobinopathies in Sub-Saharan Africa, from 2000 to 2021.**

| Region | 2000 | 2001 | 2002 | 2003 | 2004 | 2005 | 2006 | 2007 | 2008 | 2009 |
|---|---|---|---|---|---|---|---|---|---|---|
| WHO Africa Region | 5.23 (3.71 - 7.90) | 5.19 (3.63 - 7.96) | 5.17 (3.67 - 7.85) | 5.13 (3.60 - 7.88) | 5.08 (3.62 - 7.87) | 5.03 (3.56 - 7.84) | 5.01 (3.53 - 7.83) | 4.95 (3.53 - 7.72) | 4.87 (3.42 - 7.64) | 4.84 (3.39 - 7.55) |
| Western Sub-Saharan Africa | 8.76 (5.61 - 12.47) | 8.67 (5.56 - 12.19) | 8.58 (5.63 - 12.43) | 8.44 (5.56 - 12.00) | 8.31 (5.50 - 11.89) | 8.19 (5.46 - 11.45) | 8.13 (5.38 - 11.36) | 8.00 (5.29 - 11.40) | 7.81 (5.18 - 10.90) | 7.76 (5.16 - 10.83) |
| Central Sub-Saharan Africa | 7.89 (1.75 - 28.13) | 7.92 (1.74 - 28.68) | 7.94 (1.70 - 28.69) | 8.04 (1.72 - 29.39) | 8.00 (1.71 - 29.75) | 7.96 (1.69 - 29.83) | 7.96 (1.68 - 29.86) | 7.90 (1.67 - 29.26) | 7.89 (1.68 - 28.70) | 7.87 (1.69 - 27.91) |
| Eastern Sub-Saharan Africa | 2.09 (1.27 - 4.04) | 2.05 (1.24 - 3.95) | 2.04 (1.23 - 3.97) | 2.03 (1.22 - 4.06) | 2.02 (1.21 - 4.00) | 2.01 (1.20 - 4.00) | 2.00 (1.19 - 4.06) | 1.98 (1.18 - 4.00) | 1.96 (1.17 - 3.92) | 1.94 (1.15 - 3.87) |
| Southern Sub-Saharan Africa | 1.82 (1.47 - 2.20) | 1.85 (1.49 - 2.25) | 1.90 (1.57 - 2.35) | 1.95 (1.62 - 2.44) | 1.96 (1.64 - 2.49) | 1.97 (1.64 - 2.48) | 1.97 (1.64 - 2.46) | 1.95 (1.59 - 2.40) | 1.97 (1.62 - 2.42) | 1.96 (1.60 - 2.41) |
| North Africa & Middle East | 0.96 (0.81 - 1.18) | 0.94 (0.79 - 1.14) | 0.92 (0.77 - 1.11) | 0.90 (0.76 - 1.10) | 0.89 (0.74 - 1.08) | 0.87 (0.73 - 1.07) | 0.86 (0.72 - 1.05) | 0.84 (0.71 - 1.03) | 0.83 (0.70 - 1.02) | 0.82 (0.69 - 1.01) |
| Global | 1.37 (1.13 - 1.60) | 1.35 (1.10 - 1.59) | 1.34 (1.09 - 1.57) | 1.32 (1.08 - 1.56) | 1.31 (1.07 - 1.55) | 1.29 (1.05 - 1.54) | 1.28 (1.04 - 1.52) | 1.27 (1.04 - 1.51) | 1.26 (1.02 - 1.51) | 1.25 (1.02 - 1.50) |

95% UI in parentheses.

| 2010 | 2011 | 2012 | 2013 | 2014 | 2015 | 2016 | 2017 | 2018 | 2019 | 2020 | 2021 |
|---|---|---|---|---|---|---|---|---|---|---|---|
| 4.83 (3.44 - 7.50) | 4.83 (3.41 - 7.45) | 4.85 (3.44 - 7.40) | 4.89 (3.40 - 7.44) | 4.90 (3.44 - 7.47) | 4.87 (3.40 - 7.37) | 4.84 (3.39 - 7.23) | 4.76 (3.30 - 7.09) | 4.66 (3.24 - 6.98) | 4.56 (3.16 - 6.92) | 4.48 (3.00 - 6.84) | 4.40 (2.98 - 6.77) |
| 7.74 (5.16 - 10.77) | 7.75 (5.09 - 10.92) | 7.82 (5.11 - 10.79) | 7.92 (5.11 - 11.02) | 7.95 (5.14 - 11.00) | 7.88 (5.15 - 11.11) | 7.85 (5.14 - 10.82) | 7.69 (4.99 - 10.77) | 7.48 (4.91 - 10.66) | 7.27 (4.77 - 10.31) | 7.12 (4.58 - 10.21) | 6.97 (4.54 - 10.07) |
| 7.86 (1.67 - 27.77) | 7.85 (1.66 - 27.25) | 7.84 (1.68 - 27.22) | 7.85 (1.68 - 27.12) | 7.84 (1.67 - 26.62) | 7.77 (1.67 - 26.25) | 7.68 (1.64 - 25.83) | 7.58 (1.61 - 25.62) | 7.48 (1.58 - 25.59) | 7.39 (1.58 - 25.33) | 7.34 (1.58 - 25.06) | 7.25 (1.54 - 24.80) |
| 1.92 (1.14 - 3.85) | 1.91 (1.13 - 3.79) | 1.89 (1.12 - 3.73) | 1.87 (1.11 - 3.71) | 1.86 (1.10 - 3.61) | 1.84 (1.09 - 3.60) | 1.82 (1.09 - 3.57) | 1.80 (1.07 - 3.49) | 1.77 (1.06 - 3.51) | 1.75 (1.05 - 3.46) | 1.73 (1.04 - 3.42) | 1.70 (1.01 - 3.36) |
| 1.94 (1.58 - 2.38) | 1.92 (1.55 - 2.34) | 1.89 (1.52 - 2.30) | 1.84 (1.47 - 2.23) | 1.83 (1.45 - 2.20) | 1.82 (1.45 - 2.21) | 1.82 (1.43 - 2.20) | 1.77 (1.39 - 2.15) | 1.76 (1.40 - 2.17) | 1.72 (1.37 - 2.14) | 1.71 (1.37 - 2.14) | 1.69 (1.36 - 2.13) |
| 0.80 (0.68 - 0.98) | 0.78 (0.66 - 0.96) | 0.77 (0.65 - 0.94) | 0.76 (0.64 - 0.94) | 0.75 (0.64 - 0.93) | 0.74 (0.62 - 0.91) | 0.72 (0.60 - 0.89) | 0.70 (0.59 - 0.87) | 0.68 (0.57 - 0.85) | 0.67 (0.56 - 0.84) | 0.65 (0.54 - 0.82) | 0.63 (0.53 - 0.81) |
| 1.24 (1.01 - 1.48) | 1.24 (1.01 - 1.49) | 1.23 (1.00 - 1.49) | 1.24 (1.00 - 1.49) | 1.24 (1.00 - 1.50) | 1.24 (0.99 - 1.50) | 1.24 (0.99 - 1.50) | 1.22 (0.96 - 1.49) | 1.20 (0.95 - 1.48) | 1.18 (0.93 - 1.46) | 1.16 (0.89 - 1.45) | 1.15 (0.89 - 1.43) |

**Table 3. Cause of death in rate (per 100,000 population) due to haemoglobinopathies, stratified by types of HHA in the WHO African Region, from 2000 to 2021.**

| | | 2000 | | 2004 | | 2008 | |
|---|---|---|---|---|---|---|---|
| | | Crude | Age-standardized | Crude | Age-standardized | Crude | Age-standardized |
| Hemoglobinopathies* | Female | 13.39 (7.62 - 22.96) | 6.70 (4.54 - 10.78) | 13.08 (7.39 - 22.62) | 6.57 (4.37 - 10.94) | 12.65 (7.19 - 22.43) | 6.35 (4.20 - 10.61) |
| | Male | 7.20 (4.56 - 11.67) | 3.60 (2.48 - 5.81) | 6.79 (4.43 - 10.85) | 3.43 (2.37 - 5.53) | 6.30 (4.19 - 9.87) | 3.23 (2.21 - 5.15) |
| | Both | 10.47 (6.72 - 16.14) | 5.23 (3.71 - 7.89) | 10.10 (6.56 - 15.81) | 5.08 (3.62 - 7.87) | 9.63 (6.31 - 15.13) | 4.87 (3.42 - 7.64) |
| Thalassemia | Female | 1.11 (0.47 - 2.07) | 0.31 (0.17 - 0.53) | 1.06 (0.46 - 1.97) | 0.30 (0.17 - 0.48) | 1.03 (0.47 - 1.90) | 0.29 (0.17 - 0.47) |
| | Male | 0.94 (0.55 - 1.52) | 0.25 (0.17 - 0.35) | 0.87 (0.53 - 1.37) | 0.23 (0.16 - 0.32) | 0.81 (0.48 - 1.27) | 0.21 (0.15 - 0.30) |
| | Both | 1.02 (0.58 - 1.65) | 0.28 (0.19 - 0.41) | 0.96 (0.55 - 1.57) | 0.27 (0.19 - 0.38) | 0.92 (0.54 - 1.47) | 0.25 (0.18 - 0.37) |
| Thalassemia traits | Female | 6.03 (2.92 - 10.64) | 3.29 (2.26 - 4.86) | 5.79 (2.85 - 10.03) | 3.18 (2.16 - 4.71) | 5.53 (2.77 - 9.50) | 3.04 (2.09 - 4.54) |
| | Male | 3.83 (2.34 - 5.96) | 1.93 (1.31 - 2.91) | 3.56 (2.23 - 5.46) | 1.83 (1.25 - 2.78) | 3.22 (2.02 - 4.82) | 1.70 (1.17 - 2.56) |
| | Both | 4.92 (2.99 - 7.43) | 2.62 (1.93 - 3.62) | 4.66 (2.91 - 6.91) | 2.51 (1.84 - 3.54) | 4.37 (2.74 - 6.41) | 2.37 (1.74 - 3.33) |
| Sickle cell disorders | Female | 0.79 (0.37 - 1.62) | 0.60 (0.34 - 1.20) | 0.83 (0.38 - 1.78) | 0.63 (0.34 - 1.34) | 0.89 (0.42 - 1.91) | 0.66 (0.37 - 1.44) |
| | Male | 0.61 (0.28 - 1.31) | 0.50 (0.27 - 0.98) | 0.63 (0.29 - 1.32) | 0.51 (0.29 - 0.98) | 0.64 (0.30 - 1.29) | 0.51 (0.29 - 0.97) |
| | Both | 0.71 (0.36 - 1.35) | 0.55 (0.33 - 1.03) | 0.75 (0.37 - 1.42) | 0.57 (0.33 - 1.06) | 0.77 (0.40 - 1.46) | 0.59 (0.35 - 1.09) |
| Sickle cell traits | Female | 5.46 (3.23 - 10.04) | 2.50 (1.58 - 4.27) | 5.39 (3.09 - 10.36) | 2.46 (1.51 - 4.47) | 5.20 (2.90 - 10.39) | 2.36 (1.45 - 4.36) |
| | Male | 1.82 (0.92 - 4.01) | 0.92 (0.56 - 1.74) | 1.73 (0.86 - 3.83) | 0.86 (0.51 - 1.61) | 1.63 (0.81 - 3.55) | 0.81 (0.49 - 1.48) |
| | Both | 3.81 (2.32 - 6.74) | 1.78 (1.17 - 2.97) | 3.73 (2.19 - 6.85) | 1.73 (1.13 - 2.98) | 3.57 (2.11 - 6.74) | 1.66 (1.07 - 2.90) |
| G6PD Deficiency | Female | 24.27 (11.35 - 48.19) | 19.92 (12.31 - 37.05) | 24.77 (11.42 - 50.04) | 20.39 (12.29 - 40.03) | 25.72 (12.19 - 52.84) | 20.77 (12.35 - 39.77) |
| | Male | 27.26 (14.51 - 49.39) | 20.12 (12.71 - 35.87) | 27.07 (14.70 - 47.91) | 20.20 (12.83 - 35.22) | 26.36 (14.53 - 46.36) | 19.66 (12.46 - 33.46) |
| | Both | 25.95 (14.54 - 45.16) | 20.06 (13.57 - 34.32) | 26.11 (14.36 - 45.10) | 20.34 (13.36 - 35.01) | 26.24 (14.87 - 45.75) | 20.29 (13.40 - 35.51) |
| G6PD traits | Female | 0.77 (0.51 - 1.10) | 0.70 (0.47 - 1.01) | 0.77 (0.51 - 1.11) | 0.69 (0.47 - 0.98) | 0.76 (0.50 - 1.10) | 0.68 (0.46 - 0.98) |
| | Male | 0.00 (0.00 - 0.00) | 0.00 (0.00 - 0.00) | 0.00 (0.00 - 0.00) | 0.00 (0.00 - 0.00) | 0.00 (0.00 - 0.00) | 0.00 (0.00 - 0.00) |
| | Both | 0.39 (0.26 - 0.55) | 0.35 (0.24 - 0.51) | 0.39 (0.26 - 0.56) | 0.35 (0.24 - 0.49) | 0.38 (0.25 - 0.55) | 0.34 (0.23 - 0.50) |
| Other hemo-globinopathies and hemolytic anemias | Female | 188.93 (112.33 - 342.99) | 132.14 (98.27 - 178.94) | 187.48 (109.13 - 350.49) | 130.81 (95.93 - 180.61) | 181.98 (103.72 - 355.27) | 126.54 (92.74 - 175.53) |
| | Male | 110.68 (65.34 - 194.11) | 67.94 (48.97 - 93.41) | 108.72 (63.57 - 192.26) | 65.63 (47.90 - 90.51) | 101.93 (59.05 - 186.57) | 61.23 (44.09 - 85.49) |
| | Both | 152.09 (96.07 - 246.60) | 101.03 (76.08 - 131.50) | 150.36 (93.81 - 247.79) | 99.30 (74.09 - 132.52) | 144.07 (88.90 - 239.31) | 95.03 (71.84 - 127.66) |

*Hemoglobinopathies are the main HHA outcome of which the others listed in the table are subtypes.

| 2012 | | 2016 | | 2020 | | 2021 | |
|---|---|---|---|---|---|---|---|
| Crude | Age-standardized | Crude | Age-standardized | Crude | Age-standardized | Crude | Age-standardized |
| 12.66 (7.35 - 21.77) | 6.39 (4.24 - 10.34) | 12.76 (7.57 - 21.34) | 6.42 (4.33 - 10.19) | 11.57 (7.04 - 19.59) | 5.98 (3.87 - 9.79) | 11.26 (6.91 - 19.11) | 5.88 (3.76 - 9.56) |
| 6.10 (4.04 - 9.38) | 3.14 (2.14 - 4.99) | 6.02 (3.91 - 9.47) | 3.09 (2.05 - 4.87) | 5.29 (3.33 - 8.60) | 2.80 (1.80 - 4.61) | 5.08 (3.15 - 8.35) | 2.73 (1.73 - 4.55) |
| 9.53 (6.35 - 14.82) | 4.85 (3.44 - 7.40) | 9.55 (6.39 - 14.68) | 4.84 (3.39 - 7.23) | 8.59 (5.67 - 13.43) | 4.48 (3.00 - 6.84) | 8.34 (5.47 - 13.19) | 4.40 (2.98 - 6.77) |
| 1.02 (0.47 - 1.83) | 0.29 (0.17 - 0.47) | 1.04 (0.49 - 1.92) | 0.29 (0.17 - 0.46) | 0.91 (0.43 - 1.63) | 0.26 (0.15 - 0.41) | 0.85 (0.41 - 1.52) | 0.25 (0.15 - 0.40) |
| 0.78 (0.47 - 1.22) | 0.21 (0.14 - 0.29) | 0.77 (0.45 - 1.24) | 0.20 (0.13 - 0.29) | 0.66 (0.36 - 1.06) | 0.18 (0.11 - 0.26) | 0.60 (0.32 - 0.99) | 0.17 (0.10 - 0.25) |
| 0.90 (0.54 - 1.44) | 0.25 (0.17 - 0.36) | 0.91 (0.54 - 1.45) | 0.25 (0.17 - 0.36) | 0.78 (0.46 - 1.24) | 0.22 (0.15 - 0.32) | 0.72 (0.42 - 1.16) | 0.21 (0.14 - 0.30) |
| 5.69 (2.88 - 9.59) | 3.12 (2.13 - 4.64) | 5.92 (3.10 - 9.88) | 3.21 (2.21 - 4.71) | 5.02 (2.62 - 8.33) | 2.90 (1.86 - 4.33) | 4.86 (2.62 - 8.35) | 2.85 (1.87 - 4.28) |
| 3.12 (1.95 - 4.64) | 1.67 (1.13 - 2.53) | 3.11 (1.92 - 4.80) | 1.66 (1.09 - 2.55) | 2.60 (1.50 - 4.26) | 1.47 (0.92 - 2.42) | 2.51 (1.46 - 4.22) | 1.43 (0.89 - 2.39) |
| 4.39 (2.77 - 6.50) | 2.40 (1.80 - 3.28) | 4.51 (2.84 - 6.68) | 2.44 (1.76 - 3.37) | 3.80 (2.31 - 5.82) | 2.20 (1.48 - 3.22) | 3.67 (2.29 - 5.69) | 2.15 (1.44 - 3.15) |
| 0.91 (0.42 - 1.93) | 0.67 (0.37 - 1.36) | 0.87 (0.40 - 1.87) | 0.66 (0.36 - 1.32) | 0.81 (0.37 - 1.77) | 0.61 (0.33 - 1.27) | 0.80 (0.37 - 1.72) | 0.61 (0.33 - 1.27) |
| 0.63 (0.29 - 1.23) | 0.50 (0.28 - 0.93) | 0.61 (0.28 - 1.19) | 0.48 (0.27 - 0.90) | 0.54 (0.25 - 1.07) | 0.44 (0.24 - 0.82) | 0.52 (0.24 - 1.05) | 0.43 (0.23 - 0.80) |
| 0.78 (0.40 - 1.46) | 0.59 (0.35 - 1.05) | 0.75 (0.38 - 1.42) | 0.58 (0.34 - 1.04) | 0.69 (0.34 - 1.32) | 0.53 (0.30 - 0.97) | 0.68 (0.34 - 1.28) | 0.52 (0.31 - 0.96) |
| 5.05 (2.80 - 10.22) | 2.30 (1.42 - 4.12) | 4.92 (2.72 - 9.91) | 2.27 (1.42 - 3.97) | 4.83 (2.64 - 9.64) | 2.21 (1.34 - 3.88) | 4.76 (2.62 - 9.44) | 2.18 (1.33 - 3.87) |
| 1.56 (0.76 - 3.43) | 0.77 (0.46 - 1.39) | 1.53 (0.74 - 3.35) | 0.75 (0.46 - 1.33) | 1.49 (0.71 - 3.22) | 0.72 (0.44 - 1.27) | 1.44 (0.70 - 3.08) | 0.70 (0.42 - 1.24) |
| 3.46 (2.02 - 6.56) | 1.61 (1.05 - 2.77) | 3.38 (1.95 - 6.46) | 1.58 (1.03 - 2.67) | 3.32 (1.90 - 6.19) | 1.54 (1.00 - 2.54) | 3.27 (1.88 - 6.09) | 1.52 (0.98 - 2.55) |
| 26.16 (12.68 - 51.92) | 21.04 (12.73 - 40.19) | 25.27 (12.02 - 51.02) | 20.46 (12.60 - 38.83) | 22.59 (10.87 - 45.74) | 18.89 (11.17 - 37.57) | 21.99 (10.66 - 44.18) | 18.68 (11.12 - 36.94) |
| 25.74 (13.98 - 44.60) | 19.07 (11.96 - 33.17) | 25.11 (13.44 - 43.84) | 18.44 (11.24 - 31.88) | 21.49 (11.16 - 38.16) | 16.40 (9.85 - 28.58) | 20.44 (10.47 - 36.96) | 15.99 (9.41 - 28.01) |
| 26.18 (14.87 - 44.56) | 20.15 (12.99 - 33.96) | 25.43 (14.18 - 43.64) | 19.57 (12.62 - 32.86) | 22.28 (12.09 - 39.24) | 17.78 (11.22 - 31.53) | 21.47 (11.83 - 37.85) | 17.49 (10.90 - 30.61) |
| 0.74 (0.48 - 1.09) | 0.67 (0.45 - 0.96) | 0.72 (0.48 - 1.07) | 0.66 (0.44 - 0.96) | 0.71 (0.47 - 1.06) | 0.66 (0.44 - 0.95) | 0.72 (0.47 - 1.05) | 0.66 (0.44 - 0.94) |
| 0.00 (0.00 - 0.00) | 0.00 (0.00 - 0.00) | 0.00 (0.00 - 0.00) | 0.00 (0.00 - 0.00) | 0.00 (0.00 - 0.00) | 0.00 (0.00 - 0.00) | 0.00 (0.00 - 0.00) | 0.00 (0.00 - 0.00) |
| 0.37 (0.24 - 0.55) | 0.34 (0.23 - 0.49) | 0.36 (0.24 - 0.54) | 0.34 (0.23 - 0.49) | 0.36 (0.24 - 0.53) | 0.33 (0.23 - 0.48) | 0.36 (0.24 - 0.53) | 0.34 (0.22 - 0.48) |
| 181.84 (100.50 - 360.30) | 125.09 (90.57 - 176.88) | 182.84 (97.21 - 371.10) | 126.28 (92.66 - 172.62) | 182.23 (94.79 - 362.34) | 125.05 (88.20 - 178.08) | 178.82 (94.84 - 352.68) | 123.44 (88.79 - 173.27) |
| 97.70 (55.54 - 182.87) | 57.98 (41.80 - 81.00) | 96.36 (52.39 - 182.74) | 56.30 (39.57 - 79.72) | 95.00 (51.23 - 179.31) | 54.17 (37.66 - 74.56) | 91.89 (50.62 - 168.65) | 52.90 (36.71 - 73.13) |
| 141.91 (85.07 - 245.63) | 92.80 (68.90 - 125.98) | 141.86 (81.14 - 251.42) | 92.73 (68.71 - 124.80) | 141.03 (80.32 - 246.67) | 91.22 (66.40 - 122.03) | 137.93 (79.85 - 235.91) | 89.89 (65.76 - 120.91) |

**Fig 2. Regional age-standardized mortality rate attributed to HHA in the WHO Africa region from 2000–2021.**

95% UI: 1.54 - 24.80) over time. Although the number of deaths increased or remained constant across the regions of sub-Saharan Africa from 2000 to 2021, the age-standardized death rate per 100,000 population estimates decreased over the 22 years, as seen in Fig 2.

### Sex-stratified estimates

Age-standardized death rates for HHA decreased from 2000 to 2021, the female population consistently had slightly higher rates of mortality at every five-year mark although mortality decreased from 2000 to 2021 compared to their male counterparts (See Fig 3A). Furthermore, age-standardized DALYs due to HHA decreased globally from 2000 to 2021, the female population consistently had higher DALYs at every five-year mark compared to their male counterparts (see Fig 3B).

Male-to-female cause of death rate ratio decreased for HHA, G6PD deficiency, SCD and thalassemias over the 22 year span (see Fig 4A). Male-to-female cause of death rate ratio was the highest for G6PD deficiency, ranging from 0.83 to 0.70. Male-to-female incidence at birth (per 1000 births) were consistent between 2000 to 2021 in both males and females (Fig 4B). In 2021, male-to-female birth incidence per 1000 births, was 2.37 for G6PD deficiency, followed by thalassemia, thalassemia trait, SCD and SCD trait. In 2021, female HHA incidence at birth had decreased to 596.79 (95% UI 570.25 - 625.25) per 1000 births, while male HHA incidence at birth had decreased to 320.80 (299.70 - 344.96) per 1000 births (see Table 5).

### Age-stratified estimates for SCD, Thalassemia, and G6PD

The WHO African Region has a disproportionate burden of HHA among children under the age of five compared with other age groups.[5] Notably, children under age 1 and adults aged 75 years and older experience the highest burden of HHA-attributable mortality. In 2000, the all-HHA mortality rate for infants under 1 was 22.33 deaths per 100,000 population (95% UI, 14.43 – 34.15), which declined to 17.95 (95% UI, 11.64 – 28.44) by 2021. Similarly, for adults aged 75 and older, the mortality rate decreased from 25.49 (95% UI, 16.18 – 44.50) in 2000 to 21.82 (95% UI, 13.72 – 39.16) in 2021 (Table 3).

A

**Age-standardised cause of death rate (per 100,000 population) attributed to Haemoglobinopathies from 2000 to 2021 stratified by sex**

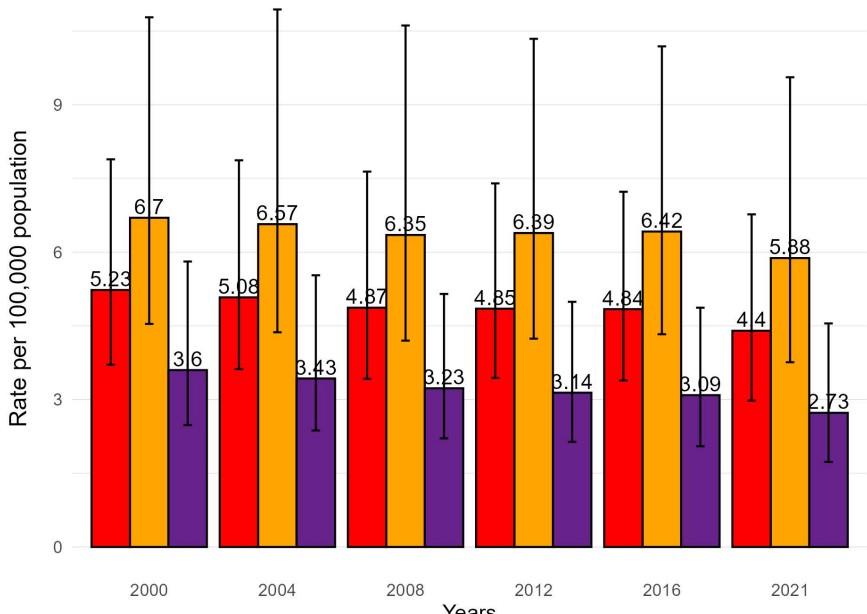

B

**Age-standardised DALYs rates (per 100,000 population) attributed to Haemoglobinopathies from 2000 to 2021 stratified by sex**

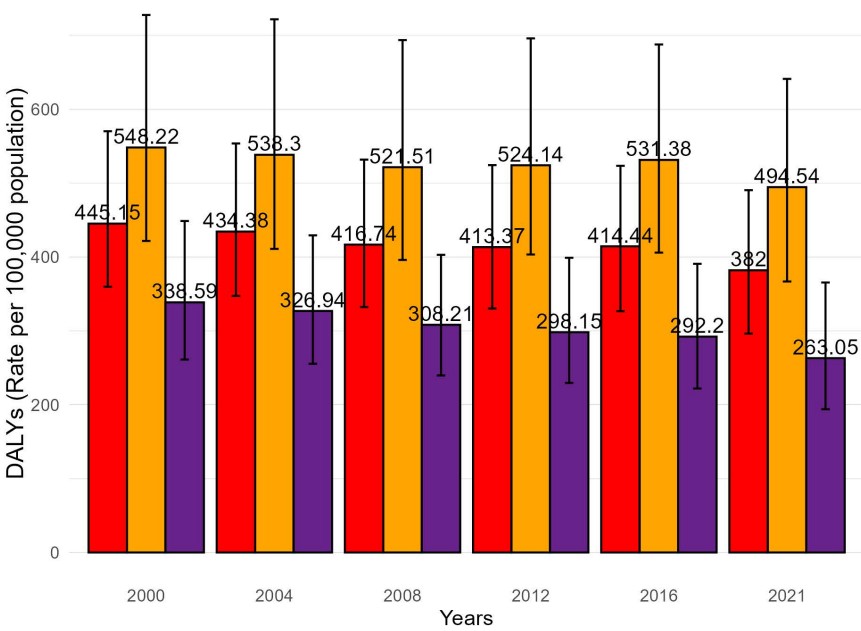

**Fig 3. Age-standardized HHA death and DALY rates by sex, 2000–2021. A**, Age-standardized cause of death rate (per 100,000 population) attributed to hemoglobinopathies from 2000–2021 stratified by sex. **B**, Age-standardized DALY rates (per 100,000 population) attributed to hemoglobinopathies from 2000–2021 stratified by sex.

**A**

**Male-to-female cause of death rate ratio by Haemoglobinopathies type in WHO Africa Region, from 2000-2021**

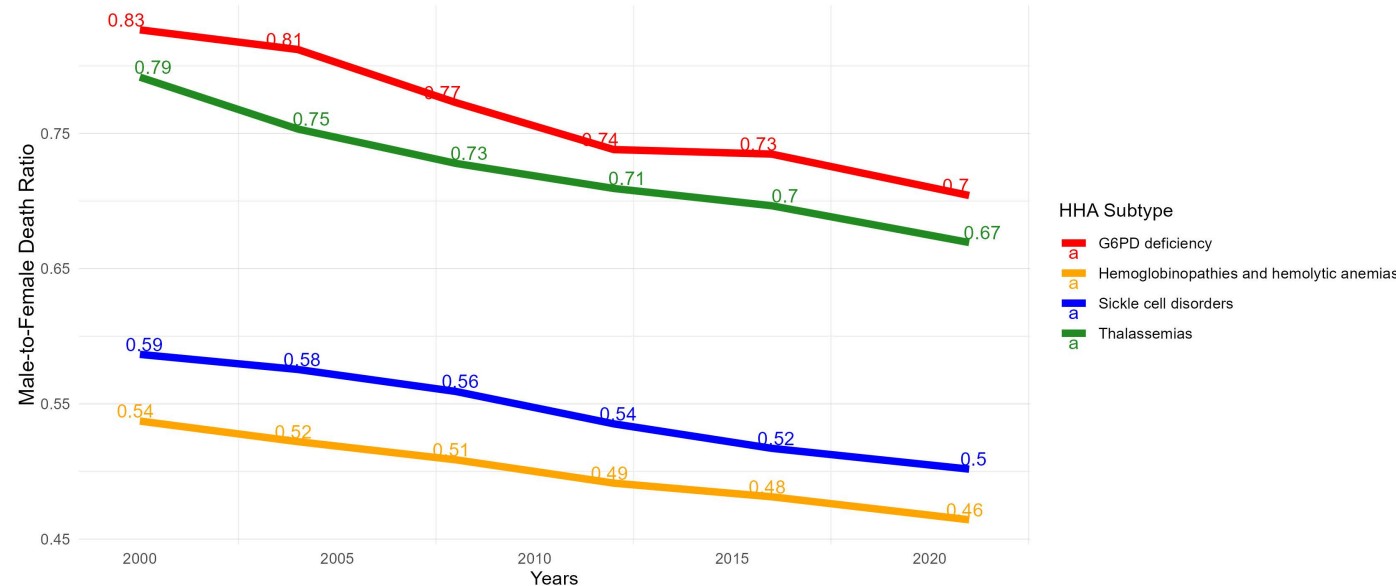

**B**

**Male-to-female birth incidence rate ratio by Haemoglobinopathies type in WHO Africa Region, from 2000 to 2021**

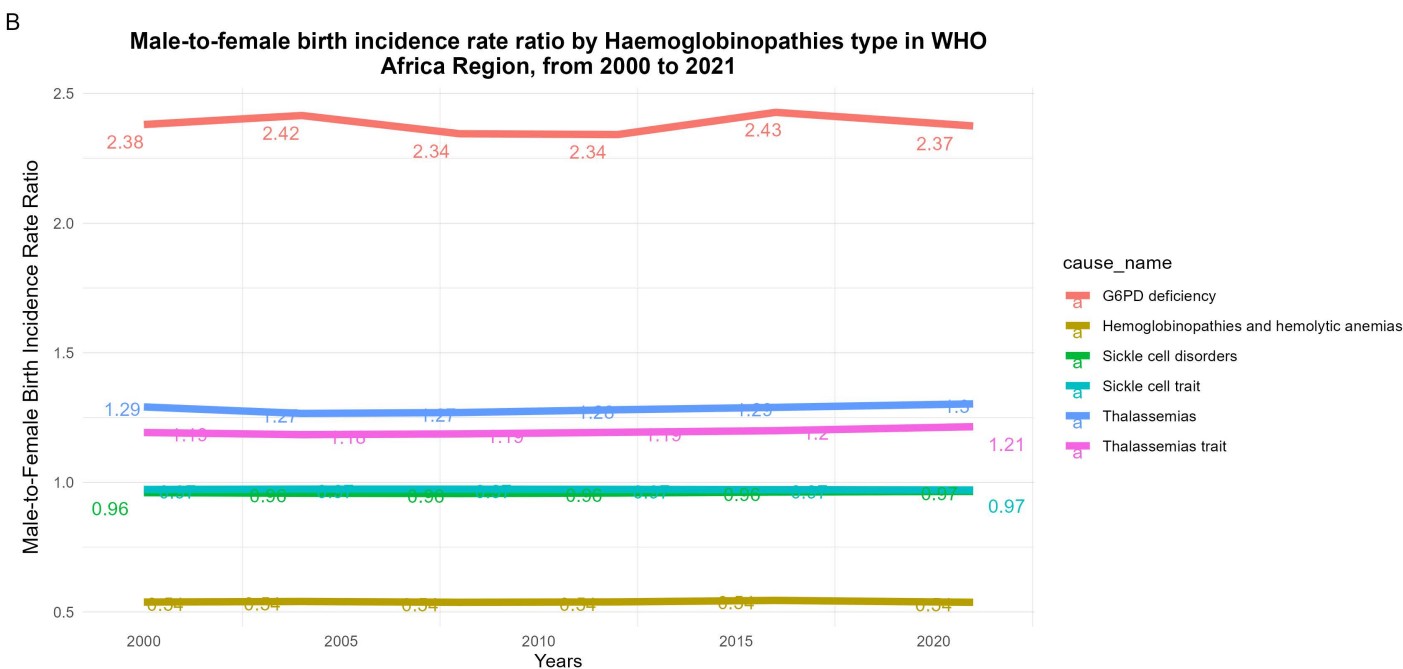

**Fig 4. Male-to-female ratios for HHA deaths and birth incidence in the WHO African Region, 2000–2021. A**, Male-to-female cause of death rate ratio by Haemoglobinopathies type in WHO Africa Region, from 2000–2021. **B**, Male-to-female birth incidence rate ratio by hemoglobinopathies type in WHO Africa Region, from 2000 to 2021.

From 2000 to 2021, cause-specific mortality rate attributed to Thalassemia, SCD and G6PD Deficiency, decreased for children under age five years. In contrast, for individuals above age five the cause-specific death rate due to HHA declined or remained constant within the different age groups from 2000 to 2021; more deaths occurred cumulatively among

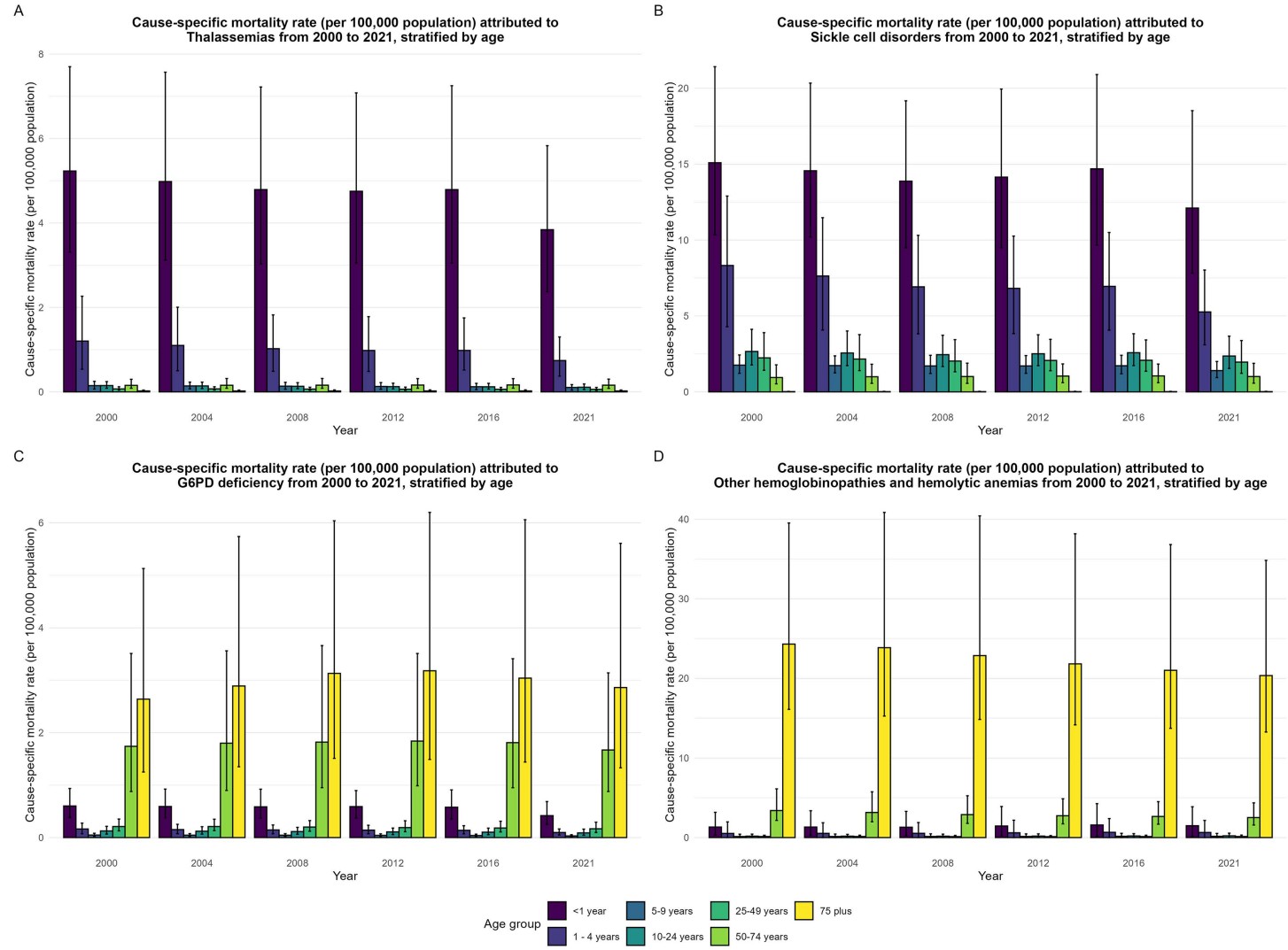

**Fig 5. Age-stratified cause-specific mortality rates for HHA in the WHO African Region, 2000–2021. A**, Cause-specific mortality rate (per 100,000 populations) attributed to thalassemias from 2000–2021, stratified by age. **B**, Cause-specific mortality rate (per 100,000 population) attributed to sickle cell disorders form 2000–2021, stratified by age. **C**, Cause-specific mortality rate (per 100,000 population) attributed to G6PD deficiency from 2000–221, stratified by age. **D**, Cause-specific mortality rate (per 100,000 population) attributed to other hemoglobinopathies and hemolytic anemias from 2000–2021, stratified by age.

children under the age 1 and between 1 and 4 years old over the last 22 years for thalassemias and SCD (Fig 5A and 5B). Cause-specific mortality within different age groups for the different types of HHA mostly saw mortality rate decline in the last 22 years or remained constant, except for ages 50–74 and over 75 years for G6PD deficiency-related deaths and over 75 years for other HHA (Fig 5C and 5D). Among HHA categories in the WHO Africa region, SCD remained the leading cause of death for all ages per 100,000 - 3.68 (95% UI: 2.04 - 6.29) (Table 3).

## SCD

Most of the deaths caused by SCD were estimated among children less than 1-year-old from 15.16 (95% UI 9.11 – 24.71) deaths per 100,000 population in 2000, decreasing to 12.18 (7.16 – 21.16) in 2021 (Table 6; Fig 5B). Furthermore, the

YLDs was greater for sickle cell trait [78.63 (52.15 – 111.38) to 74.71 (48.84 - 106.76)] for all ages than for other specific HHA categories from 2020 to 2021 respectively (Table 7). Nevertheless, YLDs declined for both SCD and hemoglobinopathies in the 22-year span (Fig 5B).

### Thalassemia, G6PD & Other HHA

Under 1 mortality for thalassemias was second to SCD (Fig 5A; Table 6). Unlike the trend observed for thalassemias and SCD, cause-specific mortality due to G6PD deficiency was highest in adults 75 years and above at 2.56 (95% UI 1.11 - 5.46) deaths per 100,000 population in 2000 and was fairly consistent to 2021 [2.75 (1.20 - 5.94)], compared to younger age groups (Fig 5C; Table 6). Likewise, people aged 75 years and more had the highest mortality rate due to other HHA of 22.90 (95% UI, 14.35 – 39.40) deaths per 100,000 population in 2000 and 19.03 (95% UI, 11.85 – 33.55) deaths per 100,000 population in 2021 (Fig 5D, Table 6).

### YLDs

Children under 1 year and those aged 1 to 4 years bore a disproportionate proportion of YLDs attributed to all HHA over the 22-year period [348.38 (235.22 - 482.76) in 2000 to 320.35 (217.06 - 451.07) in 2021 for Children under 1 and 311.93 (211.17 - 434.13) in 2000 to 278.17 (185.19 - 392.91) in 2021 for children aged 1 to 4 years] (Table 4). Furthermore, as seen in Table 4, the cause-specific YLDs rate due to HHA has been declining or remained constant within the different age groups from 2000 to 2021. Most of the YLDs for thalassemias, SCD, and G6PD occurred collectively among children under age 10 in the WHO Africa region(S1 Fig 3 – S9 Fig 9). A closer look at YLDs due to specific HHA types across different age groups, shows that infants under the age of 1 year and 1 to 4 years experienced most of the YLDs due to sickle cell trait at an average YLD rate of 118.03 (95% UI: 78.01 – 166.09) and 109.02 (95% UI: 73.18 – 153.80) per 100,000 population in 2000, respectively (Table 4), which decreased to 113.26 (95% UI: 75.08 – 157.56) and 100.76 (95% UI: 66.64 – 150.48) per 100,000 in 2021 among these groups (Table 4). Similarly, children under the age of 10 years experienced most of the YLDs due to SCD; the magnitude of YLDs for sickle cell trait was greater compared to individuals living with SCD (see S5 Fig 5 and S6 Fig 6. The trend of YLD rates due to G6PD deficiency followed a sequential downward trend across the different age groups (see S7 Fig 7 and S8 Fig 8), albeit with some slight differences in the above 10 age groups compared to sickle cell trait and SCD (Table 4).

For other HHAs, the highest burden of YLD in 2000 was observed among individuals aged 75 years and older, with an estimated rate of 94.14 (95% UI 64.92 - 134.03) YLDs per 100,000 population. By 2021, this rate declined to 79.93 (95% UI 54.09 - 114.72), reflecting a modest but notable reduction over the study period (Table 4). The second-highest YLD rates were recorded among infants under 1-year-old, with 91.96 (95% UI 62.38 - 129.58) per 100,000 population in 2000, which declined to 78.62 (95% UI 52.05 - 112.54) in 2021. Across all age groups, YLD rates due to other HHA either declined or remained constant from 2000 to 2021 (see S9 Fig 9), indicating a consistent trend of non-worsening disability burden over time. Overall, estimated YLD rates were highest for sickle cell traits compared to Thalassemia and G6PD, while YLD rates were lowest for G6PD traits (Table 4).

From 2000 to 2021, the age-standardized YLDs by HHA decreased overall from 230.80 (95% UI: 156.10 - 321.73) to 210.97 (95% UI: 141.22 – 297.19) and for all types of HHA, except for G6PD traits which remained constant. When subdivided by the main categories of HHA, sickle cell traits were the leading cause of HHA-related YLDs from 2000 to 2021 (Table 4). Thalassemias contributed the least YLDs amongst HHA types in these 22 years (seeS1 Fig 1). In 2021, sickle cell trait contributed 74.71 (95% UI 48.84 – 106.76) age-standardized YLD per 100,000 population, followed by other HHA, which caused 60.98 (95% UI 40.33 – 87.82) YLDs per 100,000 population, SCD contributed 41.08 (95% UI 26.09 – 58.61) YLDs per 100,000 population,

thalassaemia traits contributed 31.23 (95% UI: 20.42 – 44.69) YLDs per 100,000 population, G6PD deficiency contributed 1.41 (95% UI: 0.91 - 2.09) YLDs per 100,000 population, thalassemias contributed 1.20 (9 5% UI:0.74 – 1.77) YLDs per 100,000 population and G6PD traits contributed 0.36 (95% UI: 0.23 - 0.52) YLDs per 100,000 population.

## Discussion

Our findings suggest a persistent burden of HHA among populations in the WHO African Region. Although populations and birth rates have increased in the WHO African Region, with subsequent births of individuals with HHA also increasing, GBD population estimates of HHA burden in the WHO African Region highlight a slight decrease in the age-standardized death rates from 2000 to 2021 (Tables 2 and 3). While mortality due to SCD has decreased within the 22-year span, SCD has remained the leading cause of HHA-related mortality for the period (see Table 3). Other HHA (e.g., thalassemias and G6PD) contributed to fewer deaths in the 22-year span and had stable or decreasing mortality over time. Disaggregation of HHA burden indicated that the burden was highest in the Western and Central sub-Saharan Africa region during the 22-year timeframe (Table 2). Age-stratified estimated death rates indicated that mortality has been significantly higher for children under the age of 5yr or adults over 75. The HHA burden was highest at the age extremes (<1 year and 75+years), with a clear skew toward children under 1 year or adults over 75 years, depending on the HHA subtype.

Among the different types of HHA, SCD were responsible for the greatest burden of deaths between 2000 and 2021, making up approximately 1% of HHA prevalence in Africa and aligns with the disorder being the most fatal and disabling type of HHA [7,8]. Nonetheless, there has been a steady but modest declines in HHA death due to SCD during the 22 years span. While the incidence at birth of SCD has declined between 2000 and 2021, the incidence at birth of sickle cell traits peaked in 2012 before declining below 2000 levels. Of the HHA types, G6PD traits had the highest incidence at birth, while thalassemias had the lowest incidence at birth over the 22-year span. The overarching decreasing or constant incidence at birth rates observed from 2000 to 2021 is indicative of either a growing awareness of HHA through health education and genetic counseling to prevent pregnancies between HHA genetic carrier couples [32]. This decrease in or unchanging incidence at birth rates for HHA could also be indicative of the growing adoption of regional newborn screening (NBS) (i.e., state level) for HHA and/or increased awareness and promotion of genetic counseling services. State- and district-level NBS program for SCD and traits started in the early 1990s in Africa, from Ghana, [33,34] and these program have been gradually adopted by different national program across the continent, albeit challenges to program sustainability [35]. Though there was no data available for the incidence at birth for other HHA, this category of HHA was the second leading cause of death (age-standardized death rate) and disability (YLD rate) in the WHO African Region.

Regional estimates also show that Western sub-Saharan Africa bore the greatest burden of HHA on the continent from 2000 to 2021 with SCD being the most prevalent in this region. More specifically, Nigeria, has the highest prevalence of SCD globally [36]. Several biological, and environmental factors contribute to the disproportionately high burden of SCD in Western sub-Saharan Africa. One of the most well-documented explanations is the evolutionary selective pressure exerted by malaria, particularly from the parasite *Plasmodium falciparum* which has led to the persistence of the sickle cell trait at higher frequencies in populations where malaria is or has historically been endemic [37–43]. In addition to malaria-driven selection other genetic and demographic factors may contribute to the widespread transmission of SCD [37]. For example, high reproductive rates in certain communities could facilitate the continued propagation of hemoglobin variants such as HbS leading to genetically diverse population with a substantial proportion of individuals who are carriers of the sickle cell trait.

Overall, cause-specific mortality rate had the highest value for SCD, where the highest mortality rates were recorded among children under 5 years of age. This pattern is consistent with global findings and reflects several interrelated

**Table 4. DALYs in rate (per 100000 population) attributed to hemoglobinopathies, stratified by types of HHA from 2000 to 2021 stratified by sex.**

| | | 2000 | | 2004 | | 2008 | |
|---|---|---|---|---|---|---|---|
| | | Crude | Age-standardized | Crude | Age-standardized | Crude | Age-standardized |
| Hemoglobinopathies | Female | 982.60 (604.10 - 1508.47) | 548.22 (421.73 - 727.61) | 958.48 (601.06 - 1452.29) | 538.30 (410.98 - 721.75) | 927.07 (591.65 - 1386.18) | 521.51 (396.03 - 693.60) |
| | Male | 693.27 (499.12 - 946.96) | 338.59 (261.13 - 448.69) | 662.53 (493.82 - 884.91) | 326.94 (255.46 - 429.28) | 617.80 (463.36 - 819.45) | 308.21 (239.64 - 402.92) |
| | Both | 839.56 (596.70 - 1139.73) | 445.15 (359.72 - 570.22) | 812.06 (589.81 - 1095.58) | 434.38 (347.24 - 553.67) | 773.85 (568.19 - 1048.15) | 416.74 (332.25 - 531.78) |
| Thalassemia | Female | 96.71 (43.22 - 182.98) | 24.16 (13.34 - 42.60) | 93.15 (41.36 - 172.80) | 23.39 (12.78 - 38.91) | 90.31 (42.59 - 165.04) | 22.53 (12.71 - 37.45) |
| | Male | 83.99 (50.09 - 135.03) | 20.31 (13.95 - 28.94) | 77.21 (47.84 - 120.94) | 18.68 (13.45 - 26.10) | 72.30 (44.11 - 110.59) | 17.45 (12.47 - 24.17) |
| | Both | 90.25 (52.90 - 148.66) | 22.23 (15.14 - 33.33) | 85.05 (49.70 - 135.86) | 21.02 (14.52 - 30.91) | 81.16 (48.43 - 127.16) | 19.98 (14.07 - 29.05) |
| Thalassemia traits | Female | 33.21 (22.21 - 46.76) | 30.33 (20.51 - 42.78) | 33.42 (22.32 - 47.02) | 30.10 (20.39 - 42.18) | 32.63 (21.80 - 45.91) | 29.18 (19.75 - 40.91) |
| | Male | 33.69 (22.48 - 47.64) | 20.67 (13.67 - 29.38) | 33.78 (22.54 - 47.80) | 20.59 (13.69 - 29.05) | 32.77 (21.76 - 46.46) | 19.70 (13.06 - 28.06) |
| | Both | 33.57 (22.54 - 47.08) | 25.62 (17.23 - 36.18) | 33.72 (22.68 - 47.26) | 25.47 (17.11 - 35.89) | 32.82 (21.96 - 46.30) | 24.58 (16.43 - 34.67) |
| Sickle cell disorders | Female | 550.37 (288.48 - 934.12) | 261.29 (190.49 - 362.65) | 529.84 (281.77 - 871.63) | 253.63 (184.36 - 345.03) | 507.53 (273.48 - 831.44) | 243.41 (175.38 - 330.66) |
| | Male | 368.17 (242.24 - 537.19) | 167.35 (125.78 - 231.00) | 345.82 (235.47 - 490.96) | 159.36 (121.15 - 215.01) | 316.12 (216.11 - 439.84) | 149.05 (112.95 - 203.60) |
| | Both | 458.14 (292.47 - 670.93) | 214.58 (168.50 - 272.21) | 436.68 (288.74 - 619.31) | 206.68 (162.21 - 264.81) | 410.67 (272.03 - 574.01) | 196.42 (153.20 - 253.37) |
| Sickle cell traits | Female | 87.93 (58.75 - 123.67) | 79.55 (53.71 - 110.67) | 88.74 (59.43 - 124.67) | 79.14 (53.43 - 109.66) | 87.75 (59.08 - 123.08) | 78.41 (53.09 - 109.31) |
| | Male | 87.21 (58.71 - 122.96) | 51.52 (34.34 - 72.86) | 86.84 (58.75 - 121.74) | 50.84 (34.02 - 71.49) | 84.14 (57.26 - 118.06) | 48.45 (32.33 - 68.30) |
| | Both | 87.41 (58.64 - 123.00) | 65.57 (44.11 - 92.45) | 87.42 (59.13 - 122.69) | 64.97 (43.68 - 91.13) | 85.34 (58.12 - 119.82) | 63.40 (42.66 - 89.28) |
| G6PD Deficiency | Female | 982.60 (604.10 - 1508.47) | 548.22 (421.73 - 727.61) | 958.48 (601.06 - 1452.29) | 538.30 (410.98 - 721.75) | 927.07 (591.65 - 1386.18) | 521.51 (396.03 - 693.60) |
| | Male | 693.27 (499.12 - 946.96) | 338.59 (261.13 - 448.69) | 662.53 (493.82 - 884.91) | 326.94 (255.46 - 429.28) | 617.80 (463.36 - 819.45) | 308.21 (239.64 - 402.92) |
| | Both | 839.56 (596.70 - 1139.73) | 445.15 (359.72 - 570.22) | 812.06 (589.81 - 1095.58) | 434.38 (347.24 - 553.67) | 773.85 (568.19 - 1048.15) | 416.74 (332.25 - 531.78) |
| G6PD traits | Female | 96.71 (43.22 - 182.98) | 24.16 (13.34 - 42.60) | 93.15 (41.36 - 172.80) | 23.39 (12.78 - 38.91) | 90.31 (42.59 - 165.04) | 22.53 (12.71 - 37.45) |
| | Male | 83.99 (50.09 - 135.03) | 20.31 (13.95 - 28.94) | 77.21 (47.84 - 120.94) | 18.68 (13.45 - 26.10) | 72.30 (44.11 - 110.59) | 17.45 (12.47 - 24.17) |
| | Both | 90.25 (52.90 - 148.66) | 22.23 (15.14 - 33.33) | 85.05 (49.70 - 135.86) | 21.02 (14.52 - 30.91) | 81.16 (48.43 - 127.16) | 19.98 (14.07 - 29.05) |
| Other hemoglobinopathies and hemolytic anemias | Female | 33.21 (22.21 - 46.76) | 30.33 (20.51 - 42.78) | 33.42 (22.32 - 47.02) | 30.10 (20.39 - 42.18) | 32.63 (21.80 - 45.91) | 29.18 (19.75 - 40.91) |
| | Male | 33.69 (22.48 - 47.64) | 20.67 (13.67 - 29.38) | 33.78 (22.54 - 47.80) | 20.59 (13.69 - 29.05) | 32.77 (21.76 - 46.46) | 19.70 (13.06 - 28.06) |
| | Both | 33.57 (22.54 - 47.08) | 25.62 (17.23 - 36.18) | 33.72 (22.68 - 47.26) | 25.47 (17.11 - 35.89) | 32.82 (21.96 - 46.30) | 24.58 (16.43 - 34.67) |

*Hemoglobinopathies are the main HHA outcome of which the others listed in the table are subtypes

| 2012 | | 2016 | | 2020 | | 2021 | |
|---|---|---|---|---|---|---|---|
| Crude | Age-standardized | Crude | Age-standardized | Crude | Age-standardized | Crude | Age-standardized |
| 934.98 (606.35 - 1393.00) | 524.14 (403.35 - 695.92) | 954.37 (631.45 - 1394.62) | 531.38 (405.93 - 687.66) | 860.84 (590.12 - 1250.71) | 501.92 (369.95 - 655.03) | 836.53 (569.22 - 1212.72) | 494.54 (366.84 - 641.07) |
| 597.06 (447.66 - 787.45) | 298.15 (229.52 - 398.90) | 589.79 (430.53 - 795.02) | 292.20 (222.00 - 390.76) | 528.03 (376.34 - 734.66) | 268.93 (199.65 - 371.35) | 510.91 (364.37 - 715.31) | 263.05 (193.93 - 365.46) |
| 767.46 (566.11 - 1038.71) | 413.37 (330.23 - 524.37) | 773.58 (568.62 - 1044.45) | 414.44 (326.64 - 523.41) | 696.43 (504.10 - 939.03) | 388.46 (298.88 - 503.89) | 675.97 (489.87 - 910.58) | 382.00 (296.21 - 490.45) |
| 89.32 (42.10 - 160.90) | 22.10 (12.81 - 37.39) | 90.78 (44.36 - 164.53) | 22.16 (12.49 - 35.79) | 79.22 (40.46 - 141.18) | 19.65 (11.40 - 30.94) | 73.76 (37.53 - 129.59) | 18.66 (11.54 - 29.47) |
| 69.86 (41.67 - 107.01) | 16.71 (11.64 - 23.13) | 69.07 (40.54 - 108.93) | 16.45 (11.15 - 23.89) | 58.62 (32.84 - 93.34) | 14.28 (9.15 - 21.16) | 53.85 (29.58 - 87.50) | 13.37 (8.48 - 19.42) |
| 79.43 (47.74 - 126.24) | 19.40 (13.45 - 28.10) | 79.75 (49.38 - 127.83) | 19.30 (13.23 - 28.46) | 68.75 (40.50 - 109.00) | 16.96 (11.28 - 24.90) | 63.65 (37.72 - 101.34) | 16.02 (10.85 - 23.18) |
| 31.78 (21.17 - 44.80) | 28.44 (19.15 - 39.95) | 31.42 (20.76 - 44.58) | 28.25 (18.91 - 39.92) | 31.56 (20.78 - 44.75) | 28.29 (18.79 - 39.80) | 31.14 (20.56 - 44.36) | 27.90 (18.40 - 39.39) |
| 31.81 (21.05 - 45.34) | 18.85 (12.30 - 27.06) | 31.45 (20.76 - 45.42) | 18.34 (11.94 - 26.57) | 31.27 (20.42 - 45.22) | 17.81 (11.51 - 26.08) | 31.33 (20.28 - 45.02) | 17.75 (11.38 - 25.61) |
| 31.93 (21.34 - 45.14) | 23.80 (15.81 - 33.77) | 31.59 (21.00 - 44.99) | 23.48 (15.56 - 33.49) | 31.59 (20.83 - 44.89) | 23.27 (15.24 - 33.24) | 31.42 (20.62 - 44.60) | 23.04 (14.98 - 32.63) |
| 518.37 (282.11 - 837.45) | 248.91 (179.37 - 344.95) | 537.34 (303.12 - 848.47) | 255.20 (188.81 - 341.30) | 458.28 (260.94 - 723.32) | 230.25 (160.25 - 313.43) | 443.60 (255.07 - 721.36) | 225.87 (161.30 - 302.69) |
| 305.68 (206.34 - 429.31) | 146.19 (107.72 - 201.42) | 303.15 (201.72 - 437.79) | 144.80 (104.05 - 201.20) | 258.17 (165.23 - 382.57) | 129.62 (89.88 - 191.76) | 249.93 (160.70 - 386.91) | 126.55 (86.48 - 187.26) |
| 410.78 (273.23 - 584.48) | 197.83 (154.95 - 251.41) | 418.86 (278.34 - 596.72) | 200.37 (153.08 - 253.94) | 357.09 (230.84 - 517.30) | 180.37 (129.95 - 243.79) | 345.68 (227.39 - 498.39) | 176.67 (128.50 - 233.52) |
| 88.08 (59.34 - 122.95) | 78.50 (53.11 - 108.82) | 88.70 (59.88 - 124.23) | 79.46 (53.79 - 110.15) | 87.91 (59.50 - 124.08) | 79.23 (53.75 - 109.64) | 87.15 (58.60 - 123.36) | 78.37 (52.92 - 108.97) |
| 82.59 (55.45 - 117.06) | 46.88 (30.98 - 66.86) | 81.33 (54.00 - 116.59) | 45.47 (30.01 - 65.41) | 78.89 (51.57 - 114.71) | 43.12 (28.32 - 62.47) | 77.26 (50.55 - 112.55) | 42.02 (27.56 - 61.12) |
| 84.47 (57.19 - 119.38) | 62.64 (42.06 - 88.35) | 83.95 (56.59 - 119.46) | 62.43 (41.93 - 88.33) | 82.16 (54.76 - 118.23) | 61.16 (41.07 - 86.58) | 80.95 (53.86 - 116.71) | 60.16 (40.23 - 85.49) |
| 934.98 (606.35 - 1393.00) | 524.14 (403.35 - 695.92) | 954.37 (631.45 - 1394.62) | 531.38 (405.93 - 687.66) | 860.84 (590.12 - 1250.71) | 501.92 (369.95 - 655.03) | 836.53 (569.22 - 1212.72) | 494.54 (366.84 - 641.07) |
| 597.06 (447.66 - 787.45) | 298.15 (229.52 - 398.90) | 589.79 (430.53 - 795.02) | 292.20 (222.00 - 390.76) | 528.03 (376.34 - 734.66) | 268.93 (199.65 - 371.35) | 510.91 (364.37 - 715.31) | 263.05 (193.93 - 365.46) |
| 767.46 (566.11 - 1038.71) | 413.37 (330.23 - 524.37) | 773.58 (568.62 - 1044.45) | 414.44 (326.64 - 523.41) | 696.43 (504.10 - 939.03) | 388.46 (298.88 - 503.89) | 675.97 (489.87 - 910.58) | 382.00 (296.21 - 490.45) |
| 89.32 (42.10 - 160.90) | 22.10 (12.81 - 37.39) | 90.78 (44.36 - 164.53) | 22.16 (12.49 - 35.79) | 79.22 (40.46 - 141.18) | 19.65 (11.40 - 30.94) | 73.76 (37.53 - 129.59) | 18.66 (11.54 - 29.47) |
| 69.86 (41.67 - 107.01) | 16.71 (11.64 - 23.13) | 69.07 (40.54 - 108.93) | 16.45 (11.15 - 23.89) | 58.62 (32.84 - 93.34) | 14.28 (9.15 - 21.16) | 53.85 (29.58 - 87.50) | 13.37 (8.48 - 19.42) |
| 79.43 (47.74 - 126.24) | 19.40 (13.45 - 28.10) | 79.75 (49.38 - 127.83) | 19.30 (13.23 - 28.46) | 68.75 (40.50 - 109.00) | 16.96 (11.28 - 24.90) | 63.65 (37.72 - 101.34) | 16.02 (10.85 - 23.18) |
| 31.78 (21.17 - 44.80) | 28.44 (19.15 - 39.95) | 31.42 (20.76 - 44.58) | 28.25 (18.91 - 39.92) | 31.56 (20.78 - 44.75) | 28.29 (18.79 - 39.80) | 31.14 (20.56 - 44.36) | 27.90 (18.40 - 39.39) |
| 31.81 (21.05 - 45.34) | 18.85 (12.30 - 27.06) | 31.45 (20.76 - 45.42) | 18.34 (11.94 - 26.57) | 31.27 (20.42 - 45.22) | 17.81 (11.51 - 26.08) | 31.33 (20.28 - 45.02) | 17.75 (11.38 - 25.61) |
| 31.93 (21.34 - 45.14) | 23.80 (15.81 - 33.77) | 31.59 (21.00 - 44.99) | 23.48 (15.56 - 33.49) | 31.59 (20.83 - 44.89) | 23.27 (15.24 - 33.24) | 31.42 (20.62 - 44.60) | 23.04 (14.98 - 32.63) |

**Table 5. Incidence-at -birth rate (per 1000 livebirths) for haemoglobinopathies, stratified by types of HHA in the WHO African Region from 2000 to 2021, stratified by sex.**

| | | 2000 | 2004 | 2008 | 2012 | 2016 | 2020 | 2021 |
|---|---|---|---|---|---|---|---|---|
| Haemoglo-binopathies | Female | 602.29 (576.55 - 628.69) | 601.43 (575.69 - 628.56) | 609.26 (583.76 - 636.38) | 611.98 (586.06 - 639.13) | 603.87 (577.52 - 631.50) | 596.89 (570.37 - 625.45) | 596.79 (570.25 - 625.25) |
| | Male | 324.30 (304.04 - 346.30) | 325.22 (305.17 - 347.05) | 327.51 (307.04 - 350.38) | 329.91 (309.03 - 353.42) | 329.07 (308.50 - 352.19) | 320.51 (299.33 - 343.87) | 320.80 (299.70 - 344.97) |
| Thalassae-mia | Female | 0.43 (0.31 - 0.57) | 0.43 (0.32 - 0.57) | 0.44 (0.32 - 0.58) | 0.44 (0.32 - 0.58) | 0.44 (0.32 - 0.58) | 0.44 (0.32 - 0.59) | 0.44 (0.32 - 0.58) |
| | Male | 0.55 (0.41 - 0.73) | 0.55 (0.40 - 0.72) | 0.56 (0.41 - 0.73) | 0.56 (0.41 - 0.74) | 0.57 (0.42 - 0.75) | 0.57 (0.42 - 0.75) | 0.58 (0.42 - 0.76) |
| Thalassae-mia traits | Female | 27.10 (23.79 - 30.65) | 27.17 (23.89 - 30.76) | 27.22 (23.98 - 30.84) | 27.24 (24.00 - 30.87) | 27.33 (24.15 - 30.93) | 27.54 (24.32 - 31.03) | 27.31 (23.86 - 31.11) |
| | Male | 32.32 (28.45 - 36.68) | 32.18 (28.18 - 36.51) | 32.32 (28.39 - 36.58) | 32.51 (28.56 - 36.74) | 32.78 (28.76 - 37.06) | 33.03 (28.95 - 37.54) | 33.18 (29.10 - 37.93) |
| Sickle cell disorders | Female | 11.48 (9.75 - 13.60) | 11.35 (9.66 - 13.41) | 11.32 (9.64 - 13.34) | 11.37 (9.69 - 13.46) | 11.38 (9.70 - 13.50) | 11.32 (9.59 - 13.49) | 11.27 (9.53 - 13.45) |
| | Male | 11.02 (9.37 - 12.97) | 10.87 (9.27 - 12.74) | 10.84 (9.24 - 12.67) | 10.91 (9.29 - 12.86) | 10.95 (9.31 - 12.91) | 10.92 (9.33 - 12.89) | 10.88 (9.21 - 12.93) |
| Sickle cell traits | Female | 138.27 (127.63 - 149.59) | 138.98 (128.38 - 150.23) | 140.51 (129.94 - 151.89) | 141.91 (131.35 - 153.28) | 142.45 (131.74 - 153.98) | 141.75 (131.07 - 153.93) | 142.02 (131.20 - 154.01) |
| | Male | 134.49 (123.78 - 145.48) | 135.37 (124.95 - 146.15) | 136.83 (126.40 - 147.45) | 138.08 (127.62 - 148.84) | 138.44 (128.07 - 149.43) | 137.55 (126.89 - 148.64) | 137.87 (127.32 - 149.16) |
| G6PD deficiency | Female | 61.30 (54.38 - 69.27) | 60.56 (53.67 - 68.91) | 62.68 (55.66 - 71.14) | 63.14 (56.03 - 71.89) | 60.28 (53.10 - 69.02) | 58.26 (51.10 - 66.56) | 58.24 (51.07 - 66.53) |
| | Male | 145.91 (131.20 - 164.04) | 146.25 (131.33 - 164.45) | 146.97 (131.26 - 166.18) | 147.85 (131.73 - 167.80) | 146.33 (130.38 - 165.32) | 138.44 (122.04 - 157.84) | 138.30 (121.91 - 157.69) |
| G6PD traits | Female | 363.71 (349.53 - 378.27) | 362.95 (348.77 - 377.93) | 367.09 (352.61 - 382.17) | 367.87 (353.04 - 383.28) | 362.00 (346.74 - 377.99) | 357.57 (342.09 - 373.47) | 357.51 (342.02 - 373.42) |
| | Male | 0.00 (0.00 - 0.00) | 0.00 (0.00 - 0.00) | 0.00 (0.00 - 0.00) | 0.00 (0.00 - 0.00) | 0.00 (0.00 - 0.00) | 0.00 (0.00 - 0.00) | 0.00 (0.00 - 0.00) |
| Other haemoglob-inopathies and haemolytic anaemias | Female | 0.00 (0.00 - 0.00) | 0.00 (0.00 - 0.00) | 0.00 (0.00 - 0.00) | 0.00 (0.00 - 0.00) | 0.00 (0.00 - 0.00) | 0.00 (0.00 - 0.00) | 0.00 (0.00 - 0.00) |
| | Male | 0.00 (0.00 - 0.00) | 0.00 (0.00 - 0.00) | 0.00 (0.00 - 0.00) | 0.00 (0.00 - 0.00) | 0.00 (0.00 - 0.00) | 0.00 (0.00 - 0.00) | 0.00 (0.00 - 0.00) |

*Hemoglobinopathies are the main HHA outcome of which the others listed in the table are subtypes.

factors. In many countries across sub-Saharan Africa, the early identification and clinical management of SCD in infants and young children remain limited due to the absence of universal newborn screening, inadequate access to prophylactic interventions (e.g., penicillin, pneumococcal vaccines), and delays in diagnosis and referral to specialized care. Despite advances in the clinical management of SCD—including the use of hydroxyurea and disease-modifying therapies—implementation remains suboptimal in many African settings, especially for pediatric populations [18,19]. The elevated mortality rates among adults aged 75 and older may be attributed to increased vulnerability due to comorbidities, and limited health system responsiveness to the chronic complications of HHA. In older adults, late-stage complications such as cardiopulmonary dysfunction, renal impairment, and cerebrovascular disease associated with lifelong hemolysis may go undiagnosed or undertreated [44–47]. Furthermore, data from regions with similar health system profiles, such as parts of South Asia [48–50] and Latin America [51–53] reflect similar structural challenges in disease surveillance, preventive care, and long-term management.

**Table 6. Cause of death in rate (per 100000 population) attributed to hemoglobinopathies, stratified by types of HHA from 2000 to 2021 stratified by age.**

| | | 2000 | 2004 | 2009 | 2014 | 2019 | 2020 | 2021 |
|---|---|---|---|---|---|---|---|---|
| **Hemoglobinopathies** | < 1 year | 22.33 (14.43 - 34.15) | 21.54 (14.08 - 33.22) | 20.64 (13.69 - 31.44) | 21.02 (13.75 - 31.82) | 21.75 (14.10 - 32.73) | 18.73 (12.24 - 29.00) | 17.95 (11.64 - 28.44) |
| | 1-4 years | 10.23 (4.65 - 17.18) | 9.43 (4.49 - 15.11) | 8.65 (4.30 - 13.61) | 8.58 (4.47 - 13.40) | 8.77 (4.78 - 13.54) | 7.22 (4.13 - 11.19) | 6.78 (3.85 - 10.57) |
| | 5-9 years | 2.09 (1.40 - 3.06) | 2.06 (1.44 - 2.97) | 2.03 (1.37 - 2.99) | 2.02 (1.35 - 3.00) | 2.04 (1.34 - 2.98) | 1.77 (1.13 - 2.56) | 1.70 (1.08 - 2.47) |
| | 10-24 years | 3.10 (1.95 - 4.97) | 2.99 (1.89 - 4.89) | 2.86 (1.81 - 4.61) | 2.92 (1.85 - 4.62) | 2.99 (1.88 - 4.77) | 2.81 (1.69 - 4.55) | 2.77 (1.68 - 4.54) |
| | 25-49 years | 2.63 (1.64 - 4.79) | 2.56 (1.58 - 4.67) | 2.40 (1.51 - 4.32) | 2.42 (1.53 - 4.24) | 2.43 (1.52 - 4.22) | 2.32 (1.39 - 4.17) | 2.30 (1.37 - 4.10) |
| | 50-74 years | 6.24 (3.82 - 12.69) | 6.09 (3.67 - 12.64) | 5.84 (3.52 - 11.97) | 5.74 (3.44 - 11.18) | 5.60 (3.38 - 10.55) | 5.29 (3.06 - 10.12) | 5.25 (3.04 - 9.97) |
| | 75 plus | 25.49 (16.18 - 44.50) | 25.34 (15.63 - 45.28) | 24.71 (14.99 - 46.65) | 23.75 (14.71 - 42.91) | 22.78 (14.16 - 40.92) | 22.08 (13.73 - 39.98) | 21.82 (13.72 - 39.16) |
| | All ages | 10.29 (6.09 - 17.32) | 9.93 (5.91 - 16.74) | 9.47 (5.69 - 16.15) | 9.38 (5.70 - 15.57) | 9.39 (5.74 - 15.41) | 8.43 (5.19 - 14.09) | 8.17 (5.03 - 13.73) |
| **Thalassemia** | < 1 year | 5.24 (2.94 - 8.55) | 4.99 (2.85 - 8.14) | 4.80 (2.75 - 7.79) | 4.76 (2.72 - 7.55) | 4.80 (2.71 - 7.93) | 4.17 (2.31 - 6.75) | 3.85 (2.10 - 6.29) |
| | 1-4 years | 1.20 (0.45 - 2.39) | 1.10 (0.43 - 2.11) | 1.03 (0.40 - 1.95) | 0.98 (0.40 - 1.86) | 0.98 (0.41 - 1.86) | 0.82 (0.33 - 1.55) | 0.74 (0.30 - 1.42) |
| | 5-9 years | 0.15 (0.06 - 0.27) | 0.14 (0.06 - 0.25) | 0.13 (0.05 - 0.24) | 0.13 (0.05 - 0.23) | 0.12 (0.04 - 0.22) | 0.11 (0.04 - 0.20) | 0.10 (0.03 - 0.20) |
| | 10-24 years | 0.15 (0.07 - 0.27) | 0.14 (0.07 - 0.25) | 0.13 (0.06 - 0.24) | 0.12 (0.06 - 0.22) | 0.12 (0.06 - 0.22) | 0.11 (0.05 - 0.21) | 0.11 (0.05 - 0.21) |
| | 25-49 years | 0.07 (0.04 - 0.13) | 0.07 (0.04 - 0.13) | 0.07 (0.03 - 0.12) | 0.06 (0.03 - 0.12) | 0.06 (0.03 - 0.12) | 0.06 (0.03 - 0.12) | 0.06 (0.03 - 0.12) |
| | 50-74 years | 0.15 (0.08 - 0.33) | 0.16 (0.08 - 0.34) | 0.16 (0.08 - 0.34) | 0.16 (0.08 - 0.33) | 0.16 (0.08 - 0.33) | 0.15 (0.07 - 0.32) | 0.15 (0.08 - 0.32) |
| | 75 plus | 0.02 (0.01 - 0.04) | 0.02 (0.01 - 0.04) | 0.02 (0.01 - 0.05) | 0.02 (0.01 - 0.05) | 0.02 (0.01 - 0.04) | 0.02 (0.01 - 0.04) | 0.02 (0.01 - 0.04) |
| | All ages | 1.02 (0.51 - 1.80) | 0.96 (0.49 - 1.67) | 0.92 (0.47 - 1.59) | 0.90 (0.47 - 1.53) | 0.91 (0.47 - 1.58) | 0.78 (0.40 - 1.34) | 0.72 (0.36 - 1.25) |
| **Sickle cell disorders** | <1 year | 15.16 (9.11 - 24.71) | 14.63 (8.95 - 23.58) | 13.95 (8.40 - 22.00) | 14.22 (8.49 - 21.94) | 14.78 (8.85 - 23.87) | 12.46 (7.26 - 20.47) | 12.18 (7.16 - 21.16) |
| | 1-4 years | 8.35 (3.68 - 14.26) | 7.66 (3.44 - 12.77) | 6.94 (3.20 - 11.48) | 6.85 (3.19 - 11.36) | 6.98 (3.44 - 11.23) | 5.62 (2.69 - 9.07) | 5.29 (2.64 - 8.77) |
| | 5-9 years | 1.76 (1.12 - 2.65) | 1.74 (1.11 - 2.56) | 1.71 (1.08 - 2.58) | 1.72 (1.06 - 2.52) | 1.73 (1.02 - 2.59) | 1.47 (0.81 - 2.21) | 1.41 (0.83 - 2.17) |
| | 10-24 years | 2.65 (1.61 - 4.39) | 2.56 (1.56 - 4.24) | 2.44 (1.51 - 3.98) | 2.51 (1.55 - 4.04) | 2.57 (1.54 - 4.08) | 2.38 (1.36 - 3.96) | 2.35 (1.37 - 3.91) |
| | 25-49 years | 2.22 (1.35 - 4.10) | 2.14 (1.30 - 4.00) | 2.02 (1.23 - 3.66) | 2.05 (1.27 - 3.64) | 2.06 (1.25 - 3.64) | 1.96 (1.14 - 3.56) | 1.95 (1.15 - 3.53) |
| | 50-74 years | 0.95 (0.49 - 2.00) | 0.99 (0.52 - 2.04) | 1.00 (0.52 - 2.06) | 1.03 (0.56 - 2.00) | 1.03 (0.53 - 2.04) | 0.99 (0.49 - 1.94) | 0.99 (0.50 - 1.96) |
| | 75 plus | 0.02 (0.01 - 0.03) | 0.01 (0.00 - 0.04) | 0.01 (0.00 - 0.04) | 0.01 (0.00 - 0.04) | 0.01 (0.00 - 0.04) | 0.01 (0.00 - 0.03) | 0.01 (0.00 - 0.04) |
| | All ages | 4.93 (2.63 - 8.30) | 4.67 (2.54 - 7.75) | 4.38 (2.39 - 7.16) | 4.41 (2.41 - 7.12) | 4.52 (2.51 - 7.34) | 3.81 (2.06 - 6.29) | 3.68 (2.04 - 6.29) |

*(Continued)*

|  |  | 2000 | 2004 | 2009 | 2014 | 2019 | 2020 | 2021 |
|---|---|---|---|---|---|---|---|---|
| **G6PD Deficiency** | <1 year | 0.60 (0.32 - 1.02) | 0.59 (0.32 - 1.00) | 0.58 (0.32 - 0.98) | 0.59 (0.31 - 0.97) | 0.58 (0.30 - 0.99) | 0.46 (0.24 - 0.82) | 0.42 (0.21 - 0.74) |
|  | 1-4 years | 0.16 (0.06 - 0.30) | 0.15 (0.06 - 0.27) | 0.15 (0.06 - 0.26) | 0.14 (0.06 - 0.25) | 0.14 (0.06 - 0.25) | 0.11 (0.04 - 0.19) | 0.10 (0.04 - 0.18) |
|  | 5-9 years | 0.05 (0.02 - 0.09) | 0.05 (0.02 - 0.08) | 0.04 (0.02 - 0.08) | 0.04 (0.02 - 0.08) | 0.04 (0.01 - 0.07) | 0.03 (0.01 - 0.06) | 0.03 (0.01 - 0.06) |
|  | 10-24 years | 0.13 (0.06 - 0.24) | 0.12 (0.06 - 0.22) | 0.11 (0.06 - 0.21) | 0.11 (0.05 - 0.20) | 0.10 (0.05 - 0.20) | 0.09 (0.04 - 0.18) | 0.09 (0.04 - 0.18) |
|  | 25-49 years | 0.21 (0.12 - 0.40) | 0.21 (0.12 - 0.39) | 0.20 (0.11 - 0.37) | 0.19 (0.11 - 0.35) | 0.18 (0.10 - 0.35) | 0.17 (0.09 - 0.32) | 0.17 (0.09 - 0.32) |
|  | 50-74 years | 1.75 (0.82 - 3.90) | 1.79 (0.84 - 3.92) | 1.82 (0.89 - 3.99) | 1.83 (0.89 - 3.82) | 1.80 (0.88 - 3.69) | 1.67 (0.80 - 3.54) | 1.65 (0.80 - 3.48) |
|  | 75 plus | 2.56 (1.11 - 5.46) | 2.81 (1.22 - 6.21) | 3.04 (1.35 - 6.62) | 3.08 (1.35 - 6.71) | 2.94 (1.27 - 6.45) | 2.77 (1.19 - 6.03) | 2.75 (1.20 - 5.94) |
|  | All ages | 0.70 (0.32 - 1.46) | 0.73 (0.34 - 1.55) | 0.76 (0.36 - 1.60) | 0.77 (0.36 - 1.58) | 0.74 (0.34 - 1.53) | 0.68 (0.31 - 1.42) | 0.66 (0.30 - 1.38) |
| **Other hemoglobinopathies and hemolytic anemias** | <1 year | 1.33 (0.16 - 3.83) | 1.33 (0.11 - 4.22) | 1.31 (0.08 - 4.11) | 1.45 (0.05 - 4.77) | 1.59 (0.04 - 5.07) | 1.64 (0.00 - 5.18) | 1.50 (0.00 - 4.87) |
|  | 1-4 years | 0.52 (0.00 - 2.50) | 0.53 (0.00 - 2.50) | 0.53 (0.00 - 2.56) | 0.60 (0.00 - 2.65) | 0.66 (0.00 - 3.02) | 0.68 (0.00 - 2.96) | 0.64 (0.00 - 2.71) |
|  | 5-9 years | 0.13 (0.00 - 0.52) | 0.14 (0.00 - 0.59) | 0.14 (0.00 - 0.60) | 0.14 (0.00 - 0.59) | 0.16 (0.00 - 0.68) | 0.17 (0.00 - 0.67) | 0.16 (0.00 - 0.66) |
|  | 10-24 years | 0.18 (0.00 - 0.50) | 0.17 (0.00 - 0.49) | 0.17 (0.00 - 0.51) | 0.18 (0.00 - 0.56) | 0.20 (0.00 - 0.56) | 0.22 (0.00 - 0.65) | 0.22 (0.00 - 0.63) |
|  | 25-49 years | 0.13 (0.02 - 0.33) | 0.13 (0.02 - 0.33) | 0.12 (0.01 - 0.32) | 0.11 (0.01 - 0.28) | 0.12 (0.01 - 0.33) | 0.13 (0.01 - 0.36) | 0.13 (0.01 - 0.35) |
|  | 50-74 years | 3.40 (2.04 - 6.62) | 3.14 (1.86 - 6.30) | 2.87 (1.66 - 5.59) | 2.71 (1.60 - 5.22) | 2.61 (1.53 - 4.76) | 2.48 (1.44 - 4.63) | 2.46 (1.42 - 4.57) |
|  | 75 plus | 22.90 (14.35 - 39.40) | 22.50 (13.78 - 39.80) | 21.64 (13.06 - 39.52) | 20.63 (12.61 - 37.85) | 19.80 (12.23 - 35.58) | 19.28 (11.97 - 34.04) | 19.03 (11.85 - 33.55) |
|  | All ages | 3.64 (2.07 - 7.02) | 3.56 (1.97 - 7.09) | 3.41 (1.85 - 6.97) | 3.31 (1.78 - 6.82) | 3.23 (1.73 - 6.63) | 3.16 (1.68 - 6.43) | 3.10 (1.66 - 6.26) |

NB: Sickle cell traits thalassaemia traits and G6PD traits do not cause deaths; as such there is no death-related data for these types of HHA.

*Hemoglobinopathies are the main HHA outcome of which the others listed in the table are subtypes.

Since 2000, the under-5 mortality rate decreased from 76 deaths per 1000 live-births to 42 deaths per 1000 live-births in 2015 and 39 deaths per 1000 live-births in 2018 [54]. To sustain gains from newborn screening, we advocate for a "screen and treat/manage" approach, which will encourage the early adoption of life-changing therapies like prophylaxis to prevent early infant infections and hydroxyurea use [55]. Pilot programs for NBS have been initiated in several African countries, but there are no national screening and preventive programs for HHA in operation as of 2018 [56] which can be impactful.

Recent advances in treating SCD and thalassemias include gene therapy, suppression of gamma-globin repressors, luspartercept for anemia in β-thalassemia, and hematopoietic stem cell transplantation [57]. However, these evidence-based gene therapies have not been scaled and will likely be out of reach for millions of people living with HHA in the WHO Africa region for several years or decades to come. From 2006 to 2013, the WHO African Region and the Eastern Mediterranean Region (AFRO/EMRO) had the lowest regional transplant rate, while bearing the highest preva-lence and burden for thalassemias that require stem cell transplantation [58]. Only 29 of 1570 transplant teams operated

**Table 7. Cause-specific YLD rates (per 100,000 population), attributed to haemoglobinopathies, stratified by types of HHA in the WHO African Region from 2000 to 2021, stratified by age.**

| | | 2000 | 2004 | 2009 | 2014 | 2019 | 2020 | 2021 |
|---|---|---|---|---|---|---|---|---|
| Hemoglobinopathies | < 1 year | 348.38 (235.22 - 482.76) | 352.98 (238.60 - 490.10) | 344.04 (233.63 - 478.48) | 330.43 (224.09 - 461.36) | 320.50 (216.60 - 449.40) | 320.71 (216.03 - 448.81) | 320.35 (217.06 - 451.07) |
| | 1-4 years | 311.93 (211.17 - 434.13) | 317.24 (215.05 - 441.07) | 306.77 (207.93 - 427.74) | 291.26 (196.12 - 406.49) | 278.72 (185.87 - 391.22) | 278.04 (185.15 - 391.62) | 278.17 (185.19 - 392.91) |
| | 5-9 years | 260.29 (171.41 - 368.20) | 265.09 (174.84 - 372.36) | 261.66 (173.81 - 368.97) | 257.85 (170.80 - 367.65) | 258.06 (169.66 - 366.51) | 258.13 (168.08 - 366.50) | 257.60 (166.92 - 367.62) |
| | 10-24 years | 185.39 (127.46 - 256.57) | 185.41 (127.27 - 257.54) | 181.83 (125.29 - 249.68) | 179.34 (122.75 - 246.94) | 179.07 (122.82 - 247.37) | 177.81 (121.37 - 245.87) | 176.84 (120.20 - 243.69) |
| | 25-49 years | 136.97 (93.04 - 192.10) | 134.80 (92.06 - 190.46) | 130.19 (89.38 - 183.38) | 126.23 (85.89 - 177.74) | 124.30 (84.30 - 174.31) | 120.98 (81.72 - 169.42) | 120.36 (81.23 - 168.71) |
| | 50-74 years | 133.67 (89.89 - 187.14) | 132.52 (89.62 - 186.38) | 128.40 (87.29 - 180.12) | 125.45 (85.10 - 176.57) | 123.80 (83.69 - 174.51) | 119.53 (80.07 - 169.10) | 118.45 (79.32 - 167.20) |
| | 75 plus | 157.82 (109.46 - 218.81) | 155.78 (108.43 - 216.17) | 151.10 (104.89 - 210.56) | 148.45 (102.98 - 207.88) | 146.77 (101.02 - 206.57) | 139.77 (95.82 - 196.30) | 137.85 (94.64 - 193.40) |
| | All ages | 230.80 (156.10 - 321.73) | 232.63 (157.61 - 324.39) | 226.35 (153.77 - 315.83) | 218.78 (147.98 - 306.39) | 213.74 (143.73 - 300.14) | 211.63 (141.67 - 297.40) | 210.97 (141.22 - 297.19) |
| Thalassemia | < 1 year | 3.52 (2.19 - 5.18) | 3.58 (2.25 - 5.29) | 3.50 (2.18 - 5.22) | 3.35 (2.11 - 4.93) | 3.24 (2.03 - 4.75) | 3.26 (2.03 - 4.83) | 3.27 (2.04 - 4.79) |
| | 1-4 years | 2.68 (1.64 - 3.98) | 2.73 (1.69 - 4.03) | 2.65 (1.64 - 3.95) | 2.51 (1.58 - 3.73) | 2.40 (1.49 - 3.60) | 2.40 (1.50 - 3.62) | 2.41 (1.48 - 3.58) |
| | 5-9 years | 1.08 (0.67 - 1.62) | 1.07 (0.67 - 1.61) | 1.04 (0.64 - 1.57) | 1.01 (0.62 - 1.53) | 1.00 (0.62 - 1.54) | 0.99 (0.62 - 1.51) | 0.98 (0.61 - 1.50) |
| | 10-24 years | 0.42 (0.27 - 0.61) | 0.41 (0.27 - 0.61) | 0.41 (0.26 - 0.59) | 0.40 (0.26 - 0.59) | 0.39 (0.25 - 0.58) | 0.38 (0.25 - 0.57) | 0.38 (0.25 - 0.57) |
| | 25-49 years | 0.11 (0.07 - 0.16) | 0.11 (0.07 - 0.16) | 0.10 (0.06 - 0.16) | 0.10 (0.06 - 0.15) | 0.10 (0.06 - 0.15) | 0.09 (0.06 - 0.14) | 0.09 (0.06 - 0.14) |
| | 50-74 years | 0.02 (0.01 - 0.03) | 0.02 (0.01 - 0.03) | 0.02 (0.01 - 0.03) | 0.02 (0.01 - 0.03) | 0.02 (0.01 - 0.03) | 0.02 (0.01 - 0.03) | 0.02 (0.01 - 0.03) |
| | 75 plus | 0.00 (0.00 - 0.00) | 0.00 (0.00 - 0.00) | 0.00 (0.00 - 0.00) | 0.00 (0.00 - 0.00) | 0.00 (0.00 - 0.00) | 0.00 (0.00 - 0.00) | 0.00 (0.00 - 0.00) |
| | All ages | 1.31 (0.81 - 1.95) | 1.33 (0.83 - 1.97) | 1.30 (0.80 - 1.93) | 1.24 (0.78 - 1.83) | 1.19 (0.74 - 1.78) | 1.20 (0.74 - 1.79) | 1.20 (0.74 - 1.77) |
| Thalassemia traits | < 1 year | 50.22 (33.88 - 70.36) | 50.46 (34.21 - 70.63) | 49.43 (33.32 - 68.76) | 48.29 (32.52 - 67.86) | 48.20 (32.21 - 68.53) | 48.67 (32.27 - 69.10) | 48.59 (32.18 - 68.31) |
| | 1-4 years | 46.03 (30.89 - 64.30) | 46.43 (31.09 - 64.84) | 45.10 (30.34 - 63.35) | 43.56 (29.04 - 61.21) | 42.86 (28.45 - 60.98) | 43.39 (28.53 - 61.80) | 43.41 (28.59 - 61.83) |
| | 5-9 years | 43.17 (28.35 - 62.78) | 43.76 (28.73 - 63.48) | 42.63 (27.64 - 61.75) | 41.60 (27.18 - 60.62) | 41.31 (26.68 - 60.84) | 41.09 (26.54 - 59.91) | 40.85 (26.10 - 59.88) |
| | 10-24 years | 24.51 (16.08 - 35.11) | 24.24 (16.02 - 34.55) | 23.24 (15.29 - 33.39) | 22.37 (14.64 - 32.18) | 22.01 (14.41 - 31.85) | 21.69 (14.06 - 31.58) | 21.43 (13.78 - 31.20) |
| | 25-49 years | 19.83 (13.21 - 28.20) | 19.24 (12.78 - 27.41) | 18.30 (12.13 - 26.16) | 17.63 (11.59 - 25.23) | 17.33 (11.37 - 25.00) | 16.99 (11.08 - 24.61) | 16.72 (10.85 - 24.24) |
| | 50-74 years | 18.44 (12.35 - 25.94) | 18.43 (12.31 - 26.18) | 17.95 (11.96 - 25.43) | 17.46 (11.56 - 24.83) | 17.15 (11.35 - 24.25) | 16.68 (11.02 - 23.85) | 16.41 (10.77 - 23.33) |
| | 75 plus | 19.39 (13.10 - 26.63) | 19.81 (13.20 - 27.33) | 19.85 (13.19 - 27.28) | 19.86 (13.30 - 27.42) | 19.74 (13.17 - 27.58) | 19.39 (12.76 - 27.26) | 19.05 (12.52 - 26.90) |
| | All ages | 33.45 (22.34 - 47.20) | 33.60 (22.43 - 47.41) | 32.70 (21.78 - 46.18) | 31.79 (21.11 - 45.07) | 31.43 (20.76 - 45.00) | 31.41 (20.60 - 44.99) | 31.23 (20.42 - 44.69) |

*(Continued)*

| | | 2000 | 2004 | 2009 | 2014 | 2019 | 2020 | 2021 |
|---|---|---|---|---|---|---|---|---|
| **Sickle cell disorders** | <1 year | 81.62 (52.38 - 113.20) | 83.34 (53.90 - 115.29) | 81.96 (52.91 - 114.03) | 78.26 (50.27 - 109.52) | 74.72 (47.60 - 106.12) | 74.17 (47.02 - 105.23) | 73.86 (46.86 - 104.50) |
| | 1-4 years | 75.90 (49.43 - 106.42) | 77.93 (50.97 - 108.32) | 76.19 (49.35 - 106.90) | 71.87 (46.00 - 100.92) | 67.52 (42.61 - 96.17) | 66.08 (41.73 - 93.94) | 65.71 (41.32 - 93.93) |
| | 5-9 years | 65.68 (42.67 - 92.35) | 66.43 (43.34 - 93.44) | 66.16 (42.85 - 93.88) | 65.31 (42.04 - 93.62) | 64.52 (40.89 - 92.58) | 63.66 (39.57 - 91.41) | 63.29 (39.57 - 91.26) |
| | 10-24 years | 44.44 (29.82 - 61.49) | 45.61 (31.24 - 62.41) | 45.70 (30.93 - 62.48) | 45.81 (30.95 - 62.85) | 45.76 (30.30 - 63.11) | 45.30 (30.17 - 62.90) | 45.13 (29.76 - 63.11) |
| | 25-49 years | 14.66 (9.81 - 21.52) | 15.56 (10.43 - 22.78) | 15.98 (10.61 - 23.34) | 15.87 (10.61 - 23.31) | 15.40 (10.18 - 22.47) | 14.71 (9.84 - 21.55) | 14.79 (9.83 - 21.80) |
| | 50-74 years | 0.17 (0.10 - 0.28) | 0.18 (0.11 - 0.30) | 0.20 (0.11 - 0.32) | 0.20 (0.12 - 0.33) | 0.20 (0.12 - 0.33) | 0.19 (0.11 - 0.32) | 0.19 (0.11 - 0.32) |
| | 75 plus | 0.00 (0.00 - 0.00) | 0.00 (0.00 - 0.00) | 0.00 (0.00 - 0.00) | 0.00 (0.00 - 0.00) | 0.00 (0.00 - 0.00) | 0.00 (0.00 - 0.00) | 0.00 (0.00 - 0.00) |
| | All ages | 44.80 (29.20 - 62.71) | 45.87 (30.12 - 63.86) | 45.30 (29.51 - 63.48) | 43.65 (28.25 - 61.43) | 41.95 (26.79 - 59.62) | 41.27 (26.27 - 58.66) | 41.08 (26.09 - 58.61) |
| **Sickle cell traits** | <1 year | 118.03 (78.01 - 166.09) | 119.22 (78.73 - 165.93) | 117.95 (79.82 - 165.32) | 115.47 (78.08 - 163.82) | 113.64 (76.22 - 161.94) | 112.80 (75.30 - 156.93) | 113.26 (75.08 - 157.56) |
| | 1-4 years | 109.02 (73.18 - 153.80) | 110.51 (74.41 - 153.95) | 107.96 (72.89 - 150.18) | 103.97 (69.80 - 145.88) | 100.55 (66.86 - 143.15) | 100.29 (65.73 - 141.44) | 100.76 (66.64 - 142.52) |
| | 5-9 years | 96.29 (60.98 - 141.83) | 99.30 (63.43 - 144.16) | 99.55 (64.49 - 144.02) | 99.16 (63.08 - 144.91) | 99.59 (62.94 - 147.48) | 99.57 (61.49 - 148.63) | 99.83 (62.62 - 150.48) |
| | 10-24 years | 61.04 (40.35 - 87.34) | 60.87 (39.94 - 86.31) | 60.02 (39.10 - 85.50) | 59.50 (38.67 - 85.27) | 59.70 (38.70 - 86.41) | 59.74 (38.52 - 86.89) | 59.74 (38.42 - 86.16) |
| | 25-49 years | 47.82 (32.48 - 67.43) | 46.62 (31.60 - 66.01) | 44.95 (30.25 - 63.67) | 43.66 (29.11 - 62.01) | 43.24 (28.52 - 61.04) | 43.02 (28.07 - 61.30) | 42.97 (27.95 - 61.22) |
| | 50-74 years | 44.70 (30.12 - 62.12) | 44.44 (29.73 - 62.38) | 44.06 (29.53 - 61.62) | 43.73 (29.02 - 61.69) | 43.53 (28.65 - 61.69) | 42.79 (27.93 - 60.52) | 42.59 (27.92 - 60.17) |
| | 75 plus | 43.09 (28.84 - 58.66) | 42.66 (28.72 - 58.70) | 41.35 (27.84 - 56.76) | 40.17 (27.28 - 55.34) | 39.31 (26.33 - 54.48) | 38.14 (25.60 - 53.48) | 37.76 (25.47 - 53.41) |
| | All ages | 78.63 (52.15 - 111.38) | 79.27 (52.62 - 111.42) | 77.98 (52.10 - 109.66) | 76.20 (50.61 - 108.10) | 75.01 (49.38 - 107.42) | 74.58 (48.54 - 106.33) | 74.71 (48.84 - 106.76) |
| **G6PD Deficiency** | <1 year | 2.56 (1.68 - 3.68) | 2.56 (1.68 - 3.68) | 2.53 (1.70 - 3.65) | 2.49 (1.68 - 3.62) | 2.44 (1.65 - 3.51) | 2.31 (1.53 - 3.31) | 2.31 (1.53 - 3.33) |
| | 1-4 years | 2.35 (1.53 - 3.40) | 2.34 (1.55 - 3.39) | 2.29 (1.49 - 3.31) | 2.24 (1.48 - 3.24) | 2.16 (1.41 - 3.12) | 2.05 (1.35 - 3.00) | 2.06 (1.35 - 3.00) |
| | 5-9 years | 2.14 (1.35 - 3.23) | 2.18 (1.41 - 3.33) | 2.15 (1.35 - 3.27) | 2.11 (1.36 - 3.18) | 2.07 (1.28 - 3.17) | 1.97 (1.21 - 3.04) | 1.97 (1.19 - 3.08) |
| | 10-24 years | 1.11 (0.72 - 1.63) | 1.11 (0.71 - 1.61) | 1.09 (0.71 - 1.60) | 1.06 (0.67 - 1.57) | 1.02 (0.65 - 1.53) | 0.95 (0.60 - 1.47) | 0.95 (0.59 - 1.45) |
| | 25-49 years | 0.61 (0.40 - 0.92) | 0.59 (0.39 - 0.89) | 0.57 (0.38 - 0.85) | 0.55 (0.35 - 0.83) | 0.51 (0.33 - 0.78) | 0.47 (0.30 - 0.72) | 0.47 (0.30 - 0.71) |
| | 50-74 years | 0.73 (0.47 - 1.05) | 0.73 (0.47 - 1.06) | 0.71 (0.47 - 1.05) | 0.69 (0.45 - 1.02) | 0.66 (0.43 - 0.98) | 0.61 (0.40 - 0.91) | 0.61 (0.39 - 0.91) |
| | 75 plus | 0.95 (0.63 - 1.38) | 0.96 (0.63 - 1.40) | 0.97 (0.63 - 1.44) | 0.96 (0.62 - 1.41) | 0.93 (0.61 - 1.36) | 0.87 (0.57 - 1.27) | 0.86 (0.56 - 1.27) |
| | All ages | 1.60 (1.04 - 2.34) | 1.60 (1.05 - 2.34) | 1.57 (1.03 - 2.31) | 1.54 (1.01 - 2.26) | 1.49 (0.97 - 2.20) | 1.41 (0.91 - 2.09) | 1.41 (0.91 - 2.09) |

*(Continued)*

**Table 7.** (Continued)

| | | 2000 | 2004 | 2009 | 2014 | 2019 | 2020 | 2021 |
|---|---|---|---|---|---|---|---|---|
| **G6PD traits** | <1 year | 0.48 (0.32 - 0.66) | 0.48 (0.33 - 0.68) | 0.47 (0.32 - 0.68) | 0.46 (0.31 - 0.67) | 0.44 (0.30 - 0.66) | 0.44 (0.29 - 0.64) | 0.44 (0.30 - 0.65) |
| | 1-4 years | 0.45 (0.30 - 0.64) | 0.45 (0.30 - 0.65) | 0.44 (0.29 - 0.64) | 0.42 (0.27 - 0.61) | 0.40 (0.26 - 0.59) | 0.39 (0.26 - 0.59) | 0.40 (0.26 - 0.57) |
| | 5-9 years | 0.46 (0.29 - 0.68) | 0.47 (0.29 - 0.71) | 0.47 (0.30 - 0.70) | 0.47 (0.29 - 0.70) | 0.47 (0.29 - 0.71) | 0.47 (0.30 - 0.72) | 0.46 (0.30 - 0.69) |
| | 10-24 years | 0.35 (0.23 - 0.50) | 0.34 (0.22 - 0.50) | 0.34 (0.22 - 0.50) | 0.34 (0.22 - 0.50) | 0.33 (0.22 - 0.49) | 0.33 (0.22 - 0.47) | 0.33 (0.22 - 0.48) |
| | 25-49 years | 0.34 (0.23 - 0.49) | 0.33 (0.22 - 0.47) | 0.32 (0.21 - 0.46) | 0.31 (0.21 - 0.45) | 0.30 (0.21 - 0.44) | 0.30 (0.20 - 0.45) | 0.31 (0.20 - 0.45) |
| | 50-74 years | 0.28 (0.19 - 0.41) | 0.28 (0.19 - 0.40) | 0.29 (0.19 - 0.41) | 0.29 (0.19 - 0.42) | 0.28 (0.19 - 0.41) | 0.28 (0.18 - 0.41) | 0.28 (0.18 - 0.40) |
| | 75 plus | 0.26 (0.18 - 0.37) | 0.26 (0.18 - 0.38) | 0.26 (0.18 - 0.38) | 0.26 (0.17 - 0.38) | 0.25 (0.17 - 0.37) | 0.25 (0.17 - 0.37) | 0.25 (0.17 - 0.37) |
| | All ages | 0.38 (0.26 - 0.55) | 0.38 (0.25 - 0.56) | 0.38 (0.25 - 0.55) | 0.37 (0.24 - 0.54) | 0.36 (0.24 - 0.54) | 0.36 (0.23 - 0.53) | 0.36 (0.23 - 0.52) |
| **Other hemoglobinopathies and hemolytic anemias** | <1 year | 91.96 (62.38 - 129.58) | 93.34 (64.33 - 132.36) | 88.19 (59.83 - 125.31) | 82.11 (55.25 - 117.07) | 77.82 (51.56 - 110.80) | 79.05 (52.49 - 112.63) | 78.62 (52.05 - 112.54) |
| | 1-4 years | 75.50 (50.00 - 107.99) | 76.84 (51.26 - 109.73) | 72.14 (48.75 - 103.29) | 66.67 (44.44 - 95.27) | 62.84 (41.35 - 89.67) | 63.43 (41.26 - 90.57) | 63.43 (41.08 - 91.12) |
| | 5-9 years | 51.47 (33.02 - 74.88) | 51.88 (33.68 - 76.42) | 49.66 (32.28 - 73.17) | 48.18 (30.89 - 72.04) | 49.12 (31.48 - 74.47) | 50.39 (31.86 - 76.02) | 50.21 (31.80 - 75.88) |
| | 10-24 years | 53.51 (36.04 - 75.53) | 52.83 (35.62 - 74.78) | 51.04 (34.92 - 71.96) | 49.87 (34.16 - 70.29) | 49.85 (33.84 - 70.49) | 49.41 (33.18 - 69.86) | 48.88 (32.72 - 69.56) |
| | 25-49 years | 53.59 (35.64 - 74.58) | 52.37 (35.17 - 73.16) | 49.95 (33.51 - 70.02) | 48.13 (32.39 - 67.99) | 47.41 (31.49 - 67.17) | 45.38 (30.25 - 65.08) | 45.01 (30.19 - 64.86) |
| | 50-74 years | 69.32 (46.96 - 97.82) | 68.43 (46.43 - 98.43) | 65.17 (44.56 - 93.20) | 63.05 (42.75 - 89.85) | 61.97 (42.37 - 88.25) | 58.96 (40.06 - 83.97) | 58.35 (39.60 - 82.75) |
| | 75 plus | 94.14 (64.92 - 134.03) | 92.08 (63.42 - 131.68) | 88.67 (61.53 - 127.06) | 87.20 (59.87 - 125.08) | 86.54 (58.47 - 123.73) | 81.12 (55.30 - 117.56) | 79.93 (54.09 - 114.72) |
| | All ages | 70.62 (47.37 - 100.30) | 70.58 (47.65 - 100.79) | 67.12 (45.52 - 95.91) | 63.98 (43.03 - 91.61) | 62.30 (41.49 - 89.28) | 61.40 (40.71 - 88.28) | 60.98 (40.33 - 87.82) |

*Hemoglobinopathies are the main HHA outcome of which the others listed in the table are subtypes.

in 12 of 68 AFRO/EMRO countries. Significant gaps exist in managing HHA due to patient challenges with therapy adoption, access, and affordability, influenced by misinformation and fear of side effects. Recent systematic reviews on evidence-based management therapies for SCD reveal existing patient-level challenges in adoption, adherence, access to, and affordability of these therapies and medications in Africa [59,60]. These challenges are also applicable to several HHA as issues of treatment adoption, adherence, access, and affordability are more systemic of patient experiences around the chronic disease management in Africa [61]. The barriers are also influenced to an extent by misinformation, lack of awareness of the benefits of therapies, and fear of medication side effects [62]. Challenges of preventing and managing HHA also exist at the provider level in the form of lack of information on therapeutic advancements for SCD and thalassemias, reluctance to administer certain medications like hydroxyurea, limited specialist training, low numbers of trained hematologists and sickle-cell disease specialists and a lag in stem cell transplantations on the continent [58,63–73]. There is a need to build capacity at both the system level and the provider level through the availability of low-cost therapies, decentralized health maintenance of HHA patients through primary care networks, accelerated expansion of screening services, and tailored training of providers in the use and benefits of HHA therapies to increase prescription

practices [74,75]. However, general measures including implementation of national childhood immunizations schemes [76], maternal and child health programs [77–79], and health systems strengthening strategies [80,81] have benefited individuals with HHA. Furthermore, universal newborn screening could save up to 9.8 million newborns with SCD in sub-Saharan Africa [36].

The main limitation of using GBD estimates as it pertains to this study is the persistent shortage of data from several African countries, which limits the applicability of the estimates derived from the modelling data [82]. Because of these limitations, the present study presents regional data, which would be more accurate compared to country level data for recent estimates of the HHA burden in the WHO African Region for 2000–2021. As these results are likely an underestimation of HHA burden on the continent, access to primary data-driven studies from these countries would provide more reliable estimates of HHA burden. There is a need for up-to-date national databases on HHA burden, to better inform and tailor policy, research, and practice interventions for the WHO African Region. Additional innate measurement issues also persist, specifically in the ascertainment of the cause of death, which is easily influenced by the autopsy methods used. In Africa, physician-certified verbal autopsy is widely used but can be highly unreliable, depending on the level of healthcare facilities and physicians' knowledge of the disease [83].

## Conclusion

Our analysis of GBD data highlights the continued and significant burden of HHA in the WHO African Region. Between 2000 and 2021, there was a modest decline in the overall HHA burden; however, the African continent still carries a disproportionate portion of the global HHA-related mortality and disability. The greatest concentration of this burden is among children under 1 year of age, adults 75 years and older, and females, highlighting key demographic groups that remain particularly vulnerable. The findings from this study underscore the importance of providing more accurate information on HHA within the WHO-African Region. Significant data gaps persist, limiting our ability to fully understand trends in morbidity and mortality, impeding effective policy planning and resource allocation. Nonetheless, the observed decline in mortality rates—especially among children—suggests that evidence-based interventions are having an impact where implemented, offering a promising foundation for broader efforts. Despite this progress, significant steps are needed to scale-up clinical and community interventions for SCD, the leading contributor to HHA-related mortality in the region. Proven interventions—including newborn screening, early initiation of hydroxyurea therapy, routine immunizations, and infection prophylaxis—must be scaled up and more equitably delivered. Community-level health education, including premarital and carrier counseling, can further raise awareness and support early diagnosis and prevention.

To effectively address the persistent HHA burden—particularly in high-risk populations such as children under 5 years of age—these interventions must be delivered as part of a comprehensive healthcare package. This includes rigorous screening, prompt treatment initiation, and sustained disease management, alongside culturally tailored health education programs. Strengthening health systems to support the early adoption and scale-up of these evidence-based approaches will be essential to reducing the long-term impact of HHAs in the African Region.

## Supporting information

**S1 Fig. Age-standardized years lived with disability (YLD) due to types HHA in the WHO African Region, 2000–2021.**
(TIFF)

**S2 Fig. Incidence at birth of types of HHA in the WHO African Region, 2000–2021.**
(TIFF)

**S3 Fig. Years lived with disability (YLDs) rate (per 100,000 population) due thalassemias in the WHO African Region 2000–2021, stratified by age.**
(TIFF)

**S4 Fig. Years lived with disability (YLDs) rate (per 100,000 population) due thalassemia traits in the WHO African Region 2000–2021, stratified by age.**
(TIFF)

**S5 Fig. Years lived with disability (YLDs) rate (per 100,000 population) due to sickle cell disorders in the WHO African Region 2000–2021, stratified by age.**
(TIFF)

**S6 Fig. Years lived with disability (YLDs) rate (per 100,000 population) due sickle cell traits in the WHO African Region 2000–2021, stratified by age.**
(TIFF)

**S7 Fig. Years lived with disability (YLDs) rate (per 100,000 population) due to G6PD deficiency in the WHO African Region 2000–2021, stratified by age.**
(TIFF)

**S8 Fig. Years lived with disability (YLDs) rate (per 100,000 population) due to G6PD traits in the WHO African Region 2000–2021, stratified by age.**
(TIFF)

**S9 Fig. Years lived with disability (YLDs) rate (per 100,000 population) due to other HHA in the WHO African Region 2000–2021, stratified by age.**
(TIFF)

## Author contributions

**Conceptualization:** Temitope T Ojo, Joyce Gyamfi, Emmanuel Peprah.

**Data curation:** Isaac Sunday Chukwu.

**Formal analysis:** Temitope T Ojo, Prince M Amegbor, Farha Islam, Joyce Gyamfi, Andi Mai, Carly M Malburg, Seth Christopher Yaw Appiah, Haftu Asmerom Asmerom, Adeniyi Francis Fagbamigbe, Emmanuel Peprah.

**Funding acquisition:** Seth Christopher Yaw Appiah.

**Investigation:** Farha Islam, Richard Gyan Aboagye, Osaretin Christabel Okonji.

**Methodology:** Temitope T Ojo, Prince M Amegbor, Nicholas J Kassebaum, Richard Gyan Aboagye, Haftu Asmerom Asmerom, Adeniyi Francis Fagbamigbe, Sefineh Fenta.

**Project administration:** Nicholas J Kassebaum.

**Resources:** Isaac Sunday Chukwu, Adeniyi Francis Fagbamigbe.

**Supervision:** Joyce Gyamfi.

**Validation:** Joyce Gyamfi, Nicholas J Kassebaum, Richard Gyan Aboagye, Haftu Asmerom Asmerom, Isaac Sunday Chukwu, Fitsum Wolde Demisse, Sefineh Fenta, Teferi Gebru Gebremeskel, Segun Emmanuel Ibitoye, Biruk Getahun Kibret, Osaretin Christabel Okonji.

**Visualization:** Prince M Amegbor, Farha Islam, Andi Mai, Deborah B Adenikinju, Richard Gyan Aboagye, Haftu Asmerom Asmerom, Fitsum Wolde Demisse, Adeniyi Francis Fagbamigbe, Sefineh Fenta, Teferi Gebru Gebremeskel, Osaretin Christabel Okonji, Berhanu Woldu.

**Writing – original draft:** Temitope T Ojo, Farha Islam, Joyce Gyamfi, Andi Mai, Carly M Malburg, Deborah B Adenikinju, Nicholas J Kassebaum, Fitsum Wolde Demisse, Adeniyi Francis Fagbamigbe, Teferi Gebru Gebremeskel, Osaretin Christabel Okonji, Emmanuel Peprah.

**Writing – review & editing:** Temitope T Ojo, Prince M Amegbor, Joyce Gyamfi, Shimelis Tadesse Abebe, Richard Gyan Aboagye, Ganiyu Adeniyi Amusa, Seth Christopher Yaw Appiah, Haftu Asmerom Asmerom, Isaac Sunday Chukwu, Tadesse Asmamaw Dejenie, Gashaw Dessie, Mengistie Diress, Christopher Imokhuede Esezobor, Habitu Birhan Eshetu, Adeniyi Francis Fagbamigbe, Sefineh Fenta, Teferi Gebru Gebremeskel, Segun Emmanuel Ibitoye, Robel Hussen Kabthymer, Woldeteklehaymanot Dagne Kassahun, Biruk Getahun Kibret, Osaretin Christabel Okonji, Mayowa O Owolabi, Léon Muepu M Tshilolo, Berhanu Woldu, Emmanuel Peprah.

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
