## [Decision Letter · Decision Letter 0]

5 Aug 2025

PGPH-D-24-02847

Burden of hemoglobinopathies and hemolytic anemias in the World Health Organization African Region, 2000–2021: findings from the Global Burden of Disease 2021 Study

Dear Dr. Peprah,

Thank you for submitting your manuscript to PLOS Global Public Health. After careful consideration, we feel that it has merit but does not fully meet PLOS Global Public Health’s publication criteria as it currently stands. Therefore, we invite you to submit a revised version of the manuscript that addresses the points raised during the review process.

We look forward to receiving your revised manuscript.

Kind regards,

Nnodimele Onuigbo Atulomah, PhD

Academic Editor

Journal Requirements:

1. Please provide a/amend your detailed Financial Disclosure statement. This is published with the article. It must therefore be completed in full sentences and contain the exact wording you wish to be published.

2. Please send a completed 'Competing Interests' statement, including any COIs declared by your co-authors. If you have no competing interests to declare, please state "The authors have declared that no competing interests exist". Otherwise please declare all competing interests beginning with the statement "I have read the journal's policy and the authors of this manuscript have the following competing interests:"

3. Please provide separate figure files in .tif or .eps format.

https://journals.plos.org/mentalhealth/s/figures 

https://journals.plos.org/mentalhealth/s/figures#loc-file-requirements

4. Tables should not be uploaded as individual files. Please remove these files and include the Tables in your manuscript file as editable, cell-based objects. For more information about how to format tables, see our guidelines: 

https://journals.plos.org/globalpublichealth/s/tables

Reviewers' comments:

Reviewer's Responses to Questions

**Comments to the Author**

1. Does this manuscript meet PLOS Global Public Health’s publication criteria ? Is the manuscript technically sound, and do the data support the conclusions? The manuscript must describe methodologically and ethically rigorous research with conclusions that are appropriately drawn based on the data presented.

Reviewer #1: Partly

Reviewer #2: Yes

2. Has the statistical analysis been performed appropriately and rigorously?

Reviewer #1: No

Reviewer #2: Yes

3. Have the authors made all data underlying the findings in their manuscript fully available (please refer to the Data Availability Statement at the start of the manuscript PDF file)?

Reviewer #1: Yes

Reviewer #2: Yes

4. Is the manuscript presented in an intelligible fashion and written in standard English?

Reviewer #1: Yes

Reviewer #2: Yes

5. Review Comments to the Author

Reviewer #1: Abstract

The methods included in this section do not fully reflect the detailed methodology described in the main text. If the methods mentioned in the abstract are not aligned with those presented in the main section, both clarity and transparency may be compromised.

Methods

The section is brief. It is recommended to include additional details about the methods used by the Global Burden of Disease (GBD) study to enhance reader understanding. Since GBD applies a complex methodology (e.g., Bayesian adjustments, redistribution of ill-defined causes), it is important to provide a minimal description of these key elements rather than relying solely on external references.

Results

• The statements in lines 136–137, 152–153, and 164–165 do not correspond to results; they appear to be comments that would be more appropriate in the discussion. It is suggested to relocate such sentences to the discussion section if they reflect interpretations or value judgments. The results section should focus exclusively on presenting findings objectively.

• There is an inconsistency between line 215 (78.52) and Table 4 (78.62). The correct value should be verified and used consistently across the manuscript. Maintaining consistency between the text and tables is essential for analytical credibility.

• Results should be standardized to one decimal place. It is recommended to harmonize the numerical presentation (ideally to one decimal place, unless greater precision is technically justified) throughout both text and tables.

Discussion

• In line 241, specify that the reference is to children under 5 years of age, not under 4. The age group should be corrected to ensure accuracy.

• The discussion could be enriched by comparing the findings with those from other studies and by providing explanations for the higher mortality observed in children under 5 and adults aged 75 and older. It is suggested to expand the discussion by including references from other regions or countries with similar profiles. Additionally, the findings should be interpreted in the context of structural factors such as demographic transition, access to healthcare services, and the burden of communicable versus non-communicable diseases.

Reviewer #2: The article addresses a highly prevalent health burden in the WHO Africa region, 2000-2021 (i.e. that of DALYs and mortalities due to hemoglobinopathies and hemolytic anemias or HHAs). The manuscript is generally well-written but it is necessary to review it for clarity. The authors should be commended for the effort invested in coming up with their well-researched conclusions. However, one area that the authors of the article failed to address was the prevention of HHAs. As part of the conclusions/recommendations, I will suggest the authors include a statement or two on the provision or intensified use of targeted health education/counseling services e.g. pre-marital counseling, to further raise awareness about HHAs (particularly Sickle Cell Disease) in vulnerable communities across the WHO Africa region.

6. PLOS authors have the option to publish the peer review history of their article (what does this mean? ). If published, this will include your full peer review and any attached files.

**Do you want your identity to be public for this peer review?** For information about this choice, including consent withdrawal, please see our Privacy Policy .

Reviewer #1: No

Reviewer #2: **Yes: ** Olarinmoye Ayodeji O.

---

## [Editor Report · Decision Letter 1]

29 Aug 2025

Burden of hemoglobinopathies and hemolytic anemias in the World Health Organization African Region, 2000–2021: findings from the Global Burden of Disease 2021 Study

PGPH-D-24-02847R1

Dear Dr. Peprah,

We are pleased to inform you that your manuscript 'Burden of hemoglobinopathies and hemolytic anemias in the World Health Organization African Region, 2000–2021: findings from the Global Burden of Disease 2021 Study' has been provisionally accepted for publication in PLOS Global Public Health.

Best regards,

Nnodimele Onuigbo Atulomah, PhD

Academic Editor

Thank you for taking time to address all the stated concerns of the reviewers in the first round of review of your manuscript in the second submission. Congratulations.